# LLM Layers Immediately Correct Each Other

**Arjun Patrawala**     **Jiahai Feng**     **Erik Jones**     **Jacob Steinhardt**

University of California, Berkeley
arjunpatrawala@berkeley.edu

## Abstract

Recent methods in language model interpretability employ techniques such as sparse autoencoders to decompose residual stream contributions into linear, semantically-meaningful features. Our work demonstrates that an underlying assumption of these methods—that residual stream contributions build additively upon each other—is insufficient to fully explain model behavior. Specifically, we identify the Transformer Layer Correction Mechanism (TLCM), wherein adjacent transformer layers systematically counteract each other's contributions to the residual stream. TLCM appears in 5 out of 7 major open-source model families and activates across nearly all tokens in diverse texts. To understand TLCM, we show that it emerges during pretraining, operates most strongly on punctuation and numbers, and adaptively calibrates its correction strength based on the preceding layer's output. We further show that TLCM actively corrects a small subspace and promotes other subspaces, different from standard model behavior. We advance the "propose-and-reject" hypothesis: layers may propose multiple candidate features, while subsequent layers selectively filter out inappropriate ones. Finally, we discuss how our findings help explain three persistent challenges in feature-based interpretability: why extracted features descriptions often suffer from low specificity; why feature-based interventions for model steering fail at low magnitude; why recent work finds cross-layer transcoders outperform SAEs. [1]

## 1 Introduction

Mechanistic interpretability aims to understand large language models (LLMs) by dissecting the functions of their components. This research direction has critical implications for monitoring and controlling language models [29, 42, 27, 13, 39], as well as designing architectural improvements [51, 50].

A foundational assumption in many interpretability methods is that transformer layers progressively build upon each other's contributions to enrich representations in the residual stream. This perspective has motivated numerous techniques that extract features from transformer layer outputs, including linear probes [3], the logit lens [37], sparse autoencoders (SAEs) [6], and cross-layer transcoders (CLTs) [4, 12, 40]. Similarly, analyses of factual recall often characterize successive layers as gradually augmenting entity representations with additional recalled information [16, 33, 36, 21].

In this work, we introduce the Transformer Layer Correction Mechanism (TLCM), in which adjacent transformer layers systematically reverse portions of each other's contributions. Specifically, we find that in 5 out of 7 open-weight LLM families (Llama 3, OLMo, Mistral, Gemma, and Qwen2), layer $i + 1$ consistently produces contributions that partially oppose those of layer $i$. TLCM challenges the conventional view that layer contributions primarily add information to the residual stream; instead, layers actively edit the residual stream by selectively promoting and rejecting components

---

[1]Code is available at https://github.com/arjunpat/transformer-correction

39th Conference on Neural Information Processing Systems (NeurIPS 2025).

from previous layers. This suggests that analyzing output features of each layer independently may provide an incomplete or misleading understanding of a layer's function, as these features may be subsequently corrected.

To begin, we characterize TLCM through a series of observational experiments in Section 4. First, we show that TLCM is not present at initialization but emerges gradually during pretraining, suggesting that it is a persistent characteristic of LLM training dynamics. Second, we find that TLCM fires most frequently on tokens with high contextual dependency, including numbers, dates, and punctuation. We hypothesize that TLCM is important for handling tokens with high contextual dependency. Finally, we show that TLCM is the combined effort of both attention and MLP.

In Section 5, we focus our experiments on TLCM's underlying mechanisms. First, we use causal interventions to show that TLCM is adaptive and directly dependent on the prior layer, calibrating its correction strength based on the scale of the preceding layer's output. Then, we demonstrate that TLCM systematically corrects and reinforcing specific components of the previous layer by showing an empirical relationship between the eigenvectors of the transformer layer's Jacobian and the corrected subspace. Finally, we synthesize our findings into the "propose-and-reject hypothesis", which states that models can perform enrichment though a process of proposing multiple potential features and correcting the inappropriate ones.

Our findings are helpful for understanding three persistent challenges in feature-based interpretability: (1) why extracted features descriptions often suffer from low specificity; (2) why model steering interventions require high amplification to be effective; and (3) why cross-layer transcoders outperform sparse autoencoders in recovering interpretable features.

We hope our work establishes a corpus of well-characterized phenomena that provides empirical constraints for theories of language model interpretability, and that our findings inform stronger alignment and model steering techniques.

## 2 Related Work

**Feature-based interpretability.** A common interpretability paradigm interprets the role of a layer in a neural network as the features present in its output. The linear representation hypothesis posits that neural networks represent concepts as linear features in their activation space [35, 38]. Feature-based interpretability extracts features using supervised linear probes [3] as well as other more sophisticated techniques that can be unsupervised and/or causal (e.g. nostalgebraist [37], Burns et al. [8], Geiger et al. [15], Bricken et al. [6]).

**Correction mechanisms.** Some recent works have found self-repair and correction exists in models, in which if components in large language models are ablated, later components will change their behavior to compensate [43]. Self-repair mechanisms have been discovered in varied places: ablations of attention layers with later compensation [25, 32], resilience to swapping transformer layers [25], among others. Other research has found mechanisms that are conjectured to help with self-repair, like copy suppression in which an attention head will decrease the probability of predicting a token that has already appeared in the context [31]. TLCM is more pervasive (occurring multiple times on nearly all tokens) and occurs during normal model operation, not specifically during ablations.

**Residual stream characterization.** The Iterative Inference Hypothesis proposes that transformers progressively refine their latent representations through successive layers [7]. Other work builds methods that uncovers semantically-meaningful latents found in the residual stream [5]. Another line of work highlights the process of enrichment: transformer layers construct enriched representations in the residual stream to perform next-token prediction. For example, function vectors encode input-output functions and are placed in the residual stream to induce execution [48]. Additionally, models proactively consolidate entity-related information into the residual stream before it becomes relevant for prediction [21, 26, 20, 16, 19, 44].

## 3 Background

**Transformer notation.** Large language models (LLMs) convert a token sequence into a probability distribution over subsequent tokens. Input tokens are first embedded in the $d_{\mathrm{m}}$-dimensional residual space with positional embeddings, then passed through $n$ transformer layers before being unembedded

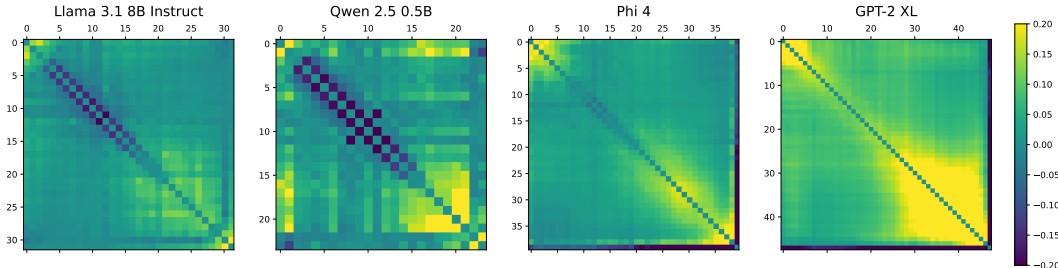

Figure 1: We plot $\mathrm{clamp}(\mathbf{M}, -0.2, 0.2)$ for four models, zeroing diagonal entries. TLCM is visible on the left two plots, characterized by off-diagonal blue stripes of negative cosine similarity, concentrated in the first two-thirds of the model. In contrast, TLCM is not present on the right two plots.

into token logits.

$$\mathbf{x}_0 = \mathrm{Embed}(\mathrm{toks}) \qquad \mathbf{x}_{i+1} = \mathrm{Layer_i}(\mathbf{x}_i) \qquad \mathrm{logits} = \mathrm{Unembed}(\mathbf{x}_n)$$

Each transformer layer contains attention and MLP sublayers that produce "contributions" to the residual stream:

$$\mathbf{x}'_i = \mathbf{x}_i + \mathrm{Attn_i}(\mathbf{x}_i) \qquad \mathbf{x}_{i+1} = \mathbf{x}'_i + \mathrm{MLP_i}(\mathbf{x}'_i)$$

where $\mathbf{x}'_i$ is an intermediate state. The marginal contribution of layer $i$ is defined as $\mathbf{c}_i = \mathrm{Attn_i}(\mathbf{x}_i) + \mathrm{MLP_i}(\mathbf{x}_i + \mathrm{Attn_i}(\mathbf{x}_i))$, making $\mathbf{x}_{i+1} = \mathbf{x}_i + \mathbf{c}_i$. We define $\mathbf{c}_{i,t}$ to be the contribution of the $i$th layer on token $t$.

**Feature-based interpretability.** Recent interpretability methods decompose residual stream contributions using sparse autoencoders (SAEs) to extract meaningful features for a residual stream contribution $\mathbf{c}$ [11, 14, 47]:

$$\mathbf{c} = \mathbf{e} + \sum_{i=1}^{k} \beta_i \mathbf{v}_i$$

where $\mathbf{e}$ is the error, $k$ is a small integer (usually less than 500), and $\mathbf{v}_i$ are the orthogonal feature vectors with activations $\beta_i$. These features can be labeled with semantic meanings and manipulated at inference time to steer model behavior. [30, 47, 9].

## 4 Transformer Layer Correction Mechanism

In this section, we introduce the Transformer Layer Correction Mechanism and conduct a series of observational studies to characterize its functioning. We first define it (Section 4.1), show that it develops during training (Section 4.2), demonstrate how it varies across token types (Section 4.3), and finally demonstrate it occurs via collaboration of both attention and MLP sublayers (Section 4.4).

### 4.1 Uncovering the Traces

We motivate the discovery of TLCM through a hypothesis about how transformer layers collaborate to enrich the residual stream. Specifically, we hypothesize that layers operate in two distinct modes: (1) contributing novel information to the residual stream, or (2) reinforcing existing information.

To test this hypothesis, we quantify layer interactions using cosine similarity between their contributions. High positive cosine similarity indicates that two layers reinforce similar representations, near-zero similarity suggests their contributions are largely disjoint, and negative cosine similarity implies one layer *reverses* or *corrects* the other's contribution. Consistent with our hypothesis, recent work has found positive cosine similarity between layer contributions in ViTs [24].

**Setup.** We test this across several open-weight language models. For a given token $t$, we define the similarity matrix $\mathbf{M}_t$, where $\mathbf{c}_{i,t}$ represents the contribution of the $i$-th transformer layer (as defined in equation 3):

$$\mathbf{M}_t[i, j] := \mathrm{cossim}(\mathbf{c}_{i,t}, \mathbf{c}_{j,t}) \text{ for } i \neq j$$

We then average these matrices element-wise across approximately 100,000 tokens, across random documents in WikiText [34] using HuggingFace Transformers [49]: $\mathbf{M} = \frac{1}{n} \sum_t \mathbf{M}_t$. These documents include a variety of languages and code.

**Results.** Plotted in Figure 1, $\mathbf{M}$ reveals a reversing effect, captured by the following observations:

- Across this large corpus, adjacent layers (layer $i$ and $i + 1$) on average have opposing contributions, evidenced by their negative cosine similarity which averages $\approx -0.2$.
- Curiously, non-adjacent layers (layer $i$ and $i + j$, $j > 1$) **do not** on average have opposing contributions; their contributions are predominately orthogonal or positively correlated, consonant with our hypothesis above.

We term this phenomenon—the systematic partial reversal of layer $i$ by layer $i + 1$—the Transformer Layer Correction Mechanism (TLCM). We see TLCM across a diverse set of model families including Llama 3 [17], OLMo [18], Mistral [22, 23], Gemma [45, 46], and Qwen2 [52], plotted in Figure 1 and Appendix D. TLCM persists across different text types, model scales, and model categories (instruction-tuned and conversational). The correction mechanism is most pronounced in the first two-thirds of each model's layers. Notably, TLCM is absent in two prominent model families: GPT-2 [41] and the Microsoft Phi models [1, 2], as shown in Figure 1. This absence may stem from their architectures—Phi-3, while based on Llama 2, incorporates dropout blocks after MLP and attention sublayers, similar to GPT-2's design. For the purposes of this paper, we study the TLCM in Llama 3.1 8B ($d_m = 4096$).

We plot the distribution of adjacent layer cosine similarity in Figure 2b, which reveals a strongly bimodal distribution. This suggests that TLCM is a *separate mode of operation*, rather than an idiosyncrasy of normal operation. Based on this plot, we define TLCM as adjacent layer contributions with cosine similarity below $-0.1$.

**Discussion.** The presence of highly anti-correlated vectors in such high-dimensional space is striking. To provide a sense of how unlikely that the two outputs of adjacent layers exhibit TLCM (i.e. have cosine similarity less than $-0.1$), we estimate the probability of this happening under two different null hypotheses in which the outputs are sampled randomly instead.

First, suppose the two outputs are drawn independently from a standard $d$-dimensional normal distribution. Then, two random vectors in $d$-dimensional space have expected cosine similarity 0 and variance $1/d$ (Appendix B.1). Therefore, as $d$ scales to 4096 (Llama 8B 3.1), the probability that TLCM occurs for two adjacent layers is as rare as a 6-sigma event.

Second, now suppose that the two outputs are drawn independently from empirical distributions of their respective layers. Compared to the earlier normally distributed setting, this accounts for the possibility that the output space of the layers could be low rank or simply be pointing in opposing directions (see Appendix B.2). We this distribution by calculating the empirical mean and standard deviation of cosine similarities between contributions from layers $i$ and $i + 1$ across four large documents. Specifically, we sample from the same layer ranges where TLCM is most active ($4 \leq i < 20$) but on adjacent contributions from different tokens. We find that a typical instance of TLCM is approximately as rare as a 3-sigma event. Therefore, TLCM—occurring multiple times on hundreds of thousands of tokens—is exceedingly unlikely to arise from chance alone. There must be a meaningful relationship between the two layers, either through a common confounder or direct dependence. In Section 5.1, we show that this relationship is indeed a direct dependence between adjacent layers.

Intuitively, transformer layers may erase information from the residual stream for various reasons: information may be only transiently useful, or deletion may free capacity for new information. Under either explanation, information written in layer $i$ could be deleted in any layer $j$ where $j > i$. However, TLCM is more constrained: reversal occurs *predominantly* in the directly subsequent layer. This pattern is puzzling from an efficiency standpoint, as it implies written information can only be exploited by a single downstream layer before erasure.

## 4.2 TLCM Develops During Training

We next investigate how TLCM emerges over training. In particular, TLCM could be:

- An inherent characteristic of transformer architectures, emerging from their fundamental design rather than through learning.
- A transient phenomenon that arises due to, for example, unstable training dynamics.
- A mechanism developed during post-training, either by RL or SFT.

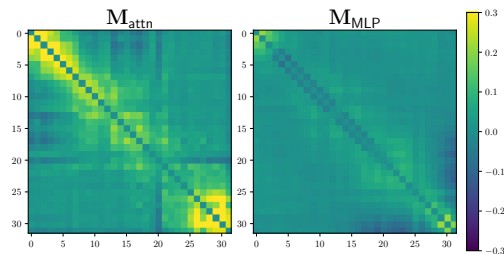 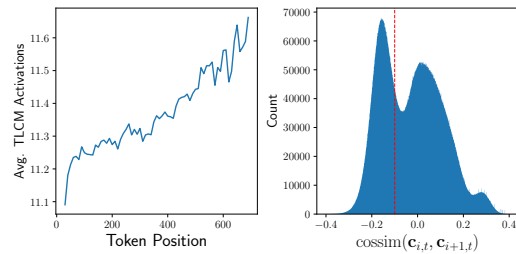

(a) Observe the correlations between attention sub-layers, plotted as $\mathrm{clip}(\mathbf{M}_{\mathrm{attn}}, -0.3, 0.3)$ on the left. Traces of TLCM are seen in the plot of $\mathrm{clip}(\mathbf{M}_{\mathrm{MLP}}, -0.3, 0.3)$ on the right.

(b) The left depicts the relationship between average TLCM activations and token position. The right illustrates the distribution of adjacent transformer layer similarities, with the dotted red line at $\mathrm{cossim} = -0.1$.

Figure 2

**Setup.** To address these possibilities, we study how TLCM develops throughout the training process. We examine seven checkpoints from each of three fully open-source models: OLMo 1B, OLMo 2 1B, and OLMo 2 7B. For each checkpoint, we compute $\mathbf{M} = \frac{1}{n}\sum_t \mathbf{M}_t$ using a standardized block of text.

**Results.** We find that TLCM emerges over the course of pretraining; in OLMo 7B, for example, it first manifests at training step 135,500 (approximately 21% through the total 651,581 steps), corresponding to roughly 0.5 trillion processed tokens. Appendix Figures A9, A10, and A11 demonstrate that the mechanism's strength increases markedly during the latter two-thirds of training for all three models. This suggests that TLCM is

- *Learning-induced*: TLCM does not appear in untrained models.
- *Persistent*: Despite weight-decay, TLCM strengthens over training and is unlikely to be some training pathology.
- *Pretraining-derived*: TLCM emerges during pretraining, not SFT or RL.

### 4.3 TLCM Varies by Token

We next study how TLCM activations vary across different token classes.

**Setup.** We curate 100 realistic queries requesting long-form content about science, technology, philosophy, government, history, and other topics. We then generate responses to these queries using Llama 3.1 8B. For each token $t$ in the response, we count the number of corrections for $t$ as: $|\{i \mid \mathrm{cossim}(\mathbf{c}_{i,t}, \mathbf{c}_{i+1,t}) < -0.1\}|$. This corpus contains 79k tokens. To ensure we identify only systematic patterns, we filter out tokens appearing fewer than 20 times and compute the mean number of TLCM activations for each unique token.

**Results.** We observe significant variation in correction frequencies across tokens on Llama 3.1 8B; some tokens had few corrections on average, while others had many more. Some qualitative examples of low-correction tokens ($< 8$ corrections per token on average) include:

> By, workshops, indigenous, vinyl, implications, cognitive, -being, innovative, innovation, regulatory, greenhouse, mindfulness, platforms, trends, therapy, example, learning, iving, planning, inclusive, cloud, classical, proposal, -friendly, sustainability, biodiversity, ting, blog, memo, system, `<|begin_of_text|>`

And some examples of high-correction tokens ($> 11$ corrections per token on average):

> 202, Jul, 26, ]\n, ,\n\n, Today, \n\n, $, Name, State, [, 201, assistant, Address, \t, 4, \n, user, at, ]\n\n, Date, , Title, ], -, Your, over, Date, %, high, ],, )\n, reach, up, share, )**, well, one, D, time, ):, take, forward, from, 1, :, access, you, 12, not, [, I, City, need, low, Thank, (, make, 10, look, your, 0, such, *

We list token-level TLCM activation statistics (mean and standard deviations) in Table A1 and A2. We also compute results for Gemma 2 2B Instruct across the same 79K tokens, listing full results in Table A3 and A4.

We find statistically significant differences in the frequency of TLCM activations. For example, the token \n averages 12.41 ($\pm$0.1 99% CI) TLCM activations while `sustainability` averages 7.63 ($\pm$0.46 99% CI) on Llama 3.1 8B. Numbers, punctuation, dates, and brackets rank highly for frequent TLCM activations, while standard English terms like `community`, `training`, and `understanding` have less frequent activations. Notably, Llama's `<|begin_of_text|>` token and Gemma's `<bos>` token exhibit the fewest average TLCM activations across the entire corpus.

**Discussion.** We observe that tokens exhibiting low TLCM correction rates typically possess unambiguous, context-independent semantic meaning. These tokens generally maintain consistent interpretation regardless of their surrounding context.

In contrast, high-correction tokens seem to demonstrate greater average contextual dependency. This category includes: 1) Numbers that form larger parts of numbers, where grasping the complete value of the number requires consolidating information *across* tokens 2) Date-related tokens which require surrounding context for complete temporal reference 3) Punctuation marks (e.g., ':') whose semantic role varies with usage context.

Recent work shows [28] that newline tokens can be used for planning upcoming tokens, a task that requires contextual processing—newline tokens and similar (\n \n, :\n, etc.) have high rates of TLCM activation. Intuitively, models may aggregate information into concluding punctuation marks because the causal attention mask prevents them from aggregating information in earlier tokens.

Llama's `<|begin_of_text|>` token and Gemma's `<bos>` are particularly revealing, with 0 and 1 TLCM activations respectively. These tokens are *always* first in the context window during training. In this position, correction is unnecessary; the optimal next-token prediction with no context is simply the empirical unigram distribution from the start of training documents, which we suspect obviates complex contextual processing.

To test this conjecture—that TLCM activations relate to contextual processing—we perform a simple check: we plot the average number of TLCM activations per token at different points in the LLM context window; Intuitively, the 1000th token can attend to 999 preceding tokens while the 5th token can only attend to 4, suggesting we should observe higher TLCM activation rates at later positions. Indeed, across 2000 long WikiText documents, we find that every 1000 tokens of additional context corresponds to approximately 1 additional TLCM activation, as demonstrated in Figure 2b.

## 4.4 Attention and MLPs Alone Do Not Explain TLCM

We next investigate the role of MLP and attention sublayers in TLCM. For instance, TLCM could arise purely from MLP-to-MLP interactions, or MLPs could selectively reverse only attention sublayer contributions. To address these questions, we conduct an analysis similar to Section 4.1, but at the sublayer level.

**Setup.** We compute a similarity matrix, instead using the attention contribution $\mathbf{M}_{\text{attn},t}[i,j] = \text{cossim}(\text{Attn}_i(\mathbf{x}_{i,t}), \text{Attn}_j(\mathbf{x}_{j,t}))$ and plot the average matrix $\mathbf{M}_{\text{attn}} = \frac{1}{n}\sum_t \mathbf{M}_{\text{attn},t}$. We also plot the average matrix for the MLPs: $\mathbf{M}_{\text{MLP}} = \frac{1}{n}\sum_t \mathbf{M}_{\text{MLP},t}$, with $\mathbf{M}_{\text{MLP},t}[i,j] = \text{cossim}(\text{MLP}_i(\mathbf{x}'_{i,t}), \text{MLP}_j(\mathbf{x}'_{j,t}))$.

**Results.** We identify three key findings. First, attention sublayers produce positively correlated contributions with other attention sublayers (Figure 2a). Second, MLP sublayers produce anti-correlated contributions with the subsequent layer's MLP, exhibiting a similar pattern to TLCM. Third, MLP contributions are also anti-correlated with the preceding attention sublayer (Appendix D). In summary, MLPs produce contributions that are anti-correlated with both the preceding MLP and attention sublayers, suggesting that MLPs are primarily responsible for *executing* corrections.

Since MLPs partially reverse contributions from both the preceding attention and MLP sublayers, the correction mechanism appears to be a transformer layer-level phenomenon. Additionally, our prior experiments suggest contextual dependency plays a role in TLCM's functioning, something only possible if TLCM operates partially through the attention sublayer. For these reasons, we focus our study of TLCM at the transformer layer level.

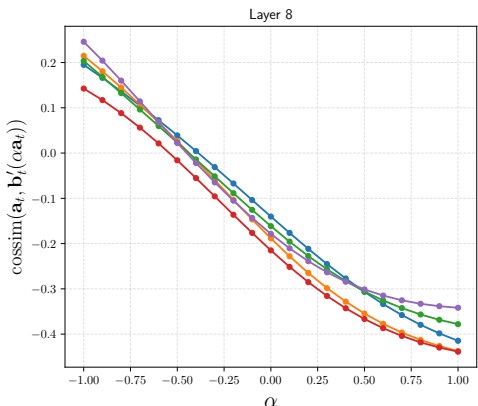

Figure 3: TLCM exhibits a linear relationship between its correction strength and the prior layer's contribution, suggesting that systematic attenuation of the prior layer, shown on 5 randomly sampled tokens. For negative $\alpha$ values, layers demonstrate compensatory behavior, evidenced by positive cosine similarity.

## 5    TLCM Adaptivity

In this section, we investigate TLCM's underlying mechanistic functioning. In Section 5.1, we show that TLCM is adaptive and directly dependent on the preceding layer's outputs. Then, in Section 5.2, we illustrate a connection between the transformer layer's Jacobian and the subspaces targeted for correction. Using this connection, we show that TLCM selectively targets subspaces for correction, while ignoring or promoting the others.

Finally, we argue for the propose-and-reject hypothesis, which explains the process of enrichment as proposing, checking, and rejecting features in a 2-layer sequence; in this picture, TLCM corrects an "undesirable" component of the previous layer.

**Notation.** For a particular token $t$, consider the marginal contribution of layer $i$ and layer $i + 1$ to be $\mathbf{a}_t$ and $\mathbf{b}_t$, respectively. Recall that the correction mechanism is currently described by a negative cosine similarity between $\mathbf{a}_t$ and $\mathbf{b}_t$ across a variety of layers and tokens. Additionally, observe that the $d_m$-dimensional input to layer $i + 1$ would be $\mathbf{x}_{i,t} = \mathbf{x}_{i-1,t} + \mathbf{a}_t$. We define a function $\mathbf{b}'_t$ that captures contribution of layer $i + 1$ when the previous layer's output is perturbed by $\boldsymbol{\Delta}$. Specifically:

$$\mathbf{b}'_t(\boldsymbol{\Delta}) := \mathrm{Layer}_{i+1}(\mathbf{x}_{i,t} + \boldsymbol{\Delta})$$

Note that $\mathbf{b}'_t(\mathbf{0}) = \mathbf{b}_t$, i.e., the original layer contribution.

### 5.1    TLCM is Adaptive

In Section 4.1, we argue that the anticorrelation between the contributions of adjacent transformer layers cannot be explained by randomness—layer $i$ and $i + 1$ anti-alignment must be explained by either a common underlying cause or a direct dependence between the layers.

Here, we perform a causal intervention to show that the anti-alignment is caused by a *direct dependence* between the layers. In particular, we find that TLCM's correction is adaptive; as we scale layer $i$'s contribution, layer $i + 1$ scales its correction, demonstrating causality.

**Setup.** To understand whether layer $i + 1$ is adaptively correcting the contribution of layer $i$, we add some perturbation $\boldsymbol{\Delta} = \alpha \mathbf{a}_t$, for $\alpha \in [-1, 1]$ to the input of layer $i + 1$. Formally, the input to layer $i + 1$ is intervened to become

$$\mathbf{x}_{i-1,t} + \mathbf{a}_t + \alpha \mathbf{a}_t = \mathbf{x}_{i-1,t} + (1 + \alpha)\mathbf{a}_t$$

If layer $i + 1$ is attuned to correcting $\mathbf{a}_t$, we should expect to see the correction increase as $\alpha$ is scaled up from $0$ and decrease as $\alpha$ is decreased from $0$. To this end, we measure the cosine similarity between $\mathbf{a}_t$ and $\mathbf{b}'_t(\alpha \mathbf{a}_t)$ at $\alpha \in \{-1, -0.9, \ldots, 0.9, 1\}$.

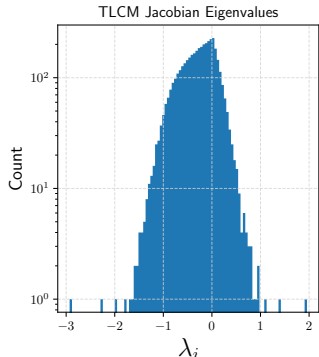
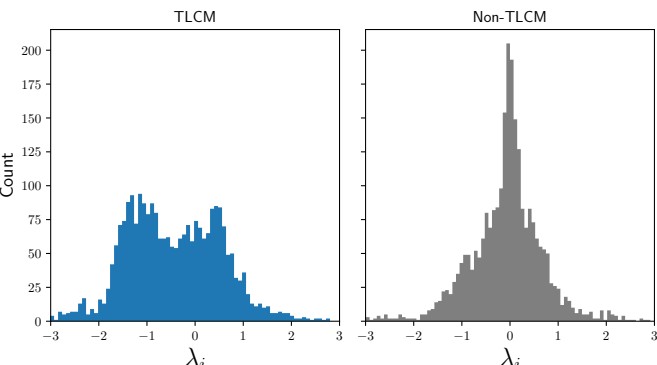

(a) Nearly 3/4 of the 4096 eigenvalues of the Jacobian $\bar{\mathbf{J}}$ are negative.

(b) Left: Strongest components of $\mathbf{a}_t$ are corrected or reinforced by the TLCM Jacobian. Right: For non-TLCM Jacobians, $\mathbf{a}_t$'s strongest components are predominantly untouched.

Figure 4

**Results.** As expected, increasing $\alpha$ leads to stronger correction. Decreasing $\alpha$ leads to less correction. Figure 3 depicts this near-linear relationship.

When $\alpha = 1$—effectively doubling $\mathbf{a}_t$ in the residual stream—layer $i + 1$ responds by adjusting its contribution to further oppose $\mathbf{a}_t$. The smooth adaptation we observe indicates layer $i + 1$'s dynamic regulation of the previous layer's contribution. This pattern appears consistently across all layers where TLCM is active, with complete results presented in Appendix E.

Our experiments also reveal a *compensatory* effect. Specifically, in Figure E at $\alpha = -1$, the cosine similarity between the adjacent layers becomes *positive*; layer $i + 1$ effectively *boosts* the contribution of layer $i$. This effect occurs most strongly in earlier layers. This finding aligns with recent research on self-repair mechanisms that enable models to recover from interventions [32]. However, we maintain cautious interpretation: extreme values of $\alpha$ place the model in counterfactual states that do not appear during standard inference or training. Moreover, this compensation effect may stem from self-repair mechanisms unrelated to TLCM.

**Discussion.** We have demonstrated that, when TLCM is active, there is a causal relationship between layer $i$ and layer $i + 1$'s correction. Adjusting layer $i$'s contribution elicits an immediate response from layer $i + 1$.

This finding is striking for a subtle reason. For layer $i + 1$ to respond sensatively to layer $i$, it must *identify* which component of the residual stream was added by layer $i$ versus components contributed by earlier layers. This is a challenging task; layer $i + 1$ would need sophisticated mechanisms to isolate layer $i$'s specific contribution. We therefore consider an alternative hypothesis: Layer $i$'s contribution might comprise both a "desirable" and "undesirable" component; layer $i + 1$ simply targets the latter for correction. For instance, layer $i$ might propose $n$ features to the residual stream, after which layer $i + 1$ verifies and removes only incorrect features while leaving the rest intact. This naturally motivates two questions:

1. Does TLCM correct the entire previous layer contribution, or does it target specific components?
2. If TLCM correction is selective, can we identify which subspaces are targeted most aggressively?

## 5.2 Isolating the Correction Subspace

In this section, we execute an experiment to answer the prior two questions. A key unknown is *how* the previous layer's contribution is attenuated: is it scaled back uniformly, or are specific subspaces selectively targeted for correction?

The layer Jacobian—the local linearization of the both the attention and MLP sublayers—contains information about how the entire layer will respond to input perturbations. By developing tools to characterize and visualize the Jacobian, we can understand which vector directions will be aggressively corrected by the layer versus which directions will remain untouched.

**Notation.** As previously defined, consider $\mathbf{b}'_t$ to be the function representing transformer layer $i+1$. We can linearize this function by computing the Jacobian $\nabla \mathbf{b}'_t(0) = \mathbf{J} \in \mathbb{R}^{d_{\text{model}} \times d_{\text{model}}}$ of the transformer layer, such that $\mathbf{b}'_t(\mathbf{\Delta}) \approx \mathbf{b}_t + \mathbf{J}\mathbf{\Delta}$. Finally, let $\overline{\mathbf{J}} := \frac{1}{2}(\mathbf{J} + \mathbf{J}^\top)$ be the symmetrized Jacobian.

**Claim**. The eigenvectors of $\overline{\mathbf{J}}$ with negative eigenvalues correspond to the corrected subspace, with the eigenvalue dictating the correction strength. In a similar vein, we argue that the positive eigenvalue eigenvectors are reinforced proportional to their eigenvalue. Finally, eigenvectors with near zero eigenvalue are mostly untouched by this layer.

**Proof.** An input perturbation to layer $i+1$ is "corrected" if $\text{cossim}(\mathbf{b}'_t(\mathbf{\Delta}) - \mathbf{b}_t, \mathbf{\Delta}) < 0$. Observe that this condition is true if and only if $\langle \mathbf{b'_t}(\mathbf{\Delta}) - \mathbf{b_t}, \mathbf{\Delta} \rangle < 0$. For a small enough perturbation, the condition is equivalent to $\langle \mathbf{J}\mathbf{\Delta}, \mathbf{\Delta} \rangle < 0$. Because $\langle \mathbf{J}\mathbf{\Delta}, \mathbf{\Delta} \rangle = \langle \mathbf{\Delta}, \mathbf{J}\mathbf{\Delta} \rangle$, the condition becomes:

$$\langle \mathbf{J}\mathbf{\Delta}, \mathbf{\Delta} \rangle = \frac{1}{2}\left(\langle \mathbf{J}\mathbf{\Delta}, \mathbf{\Delta} \rangle + \langle \mathbf{\Delta}, \mathbf{J}\mathbf{\Delta} \rangle\right) = \langle \frac{1}{2}(\mathbf{J} + \mathbf{J}^\top)\mathbf{\Delta}, \mathbf{\Delta} \rangle = \langle \overline{\mathbf{J}}\mathbf{\Delta}, \mathbf{\Delta} \rangle < 0 \qquad (1)$$

Moreover, by the spectral theorem, $\overline{\mathbf{J}}$ has an eigendecomposition $\overline{\mathbf{J}} = \mathbf{Q}\mathbf{V}\mathbf{Q}^\top$ with unitary eigenvector matrix $\mathbf{Q} \in \mathbb{R}^{d_{\text{m}} \times d_{\text{m}}}$ and eigenvalues are $\text{diag}(\mathbf{V})$. Plugging this into inequality 1:

$$\langle \overline{\mathbf{J}}\mathbf{\Delta}, \mathbf{\Delta} \rangle = \langle \mathbf{Q}\mathbf{V}\mathbf{Q}^\top\mathbf{\Delta}, \mathbf{\Delta} \rangle = \mathbf{\Delta}^\top \mathbf{Q}\mathbf{V}\mathbf{Q}^\top \mathbf{\Delta} = (\mathbf{Q}^\top\mathbf{\Delta})^\top \mathbf{V}(\mathbf{Q}^\top\mathbf{\Delta}) < 0 \qquad (2)$$

In summary, we have reduced our original condition for a corrected perturbation, $\text{cossim}(\mathbf{b}'_t(\mathbf{\Delta}) - \mathbf{b}_t, \mathbf{\Delta}) < 0$, into something more tractable to analyze: $(\mathbf{Q}^\top\mathbf{\Delta})^\top \mathbf{V}(\mathbf{Q}^\top\mathbf{\Delta}) < 0$. Observe that $\mathbf{Q}^\top\mathbf{\Delta} \in \mathbb{R}^{d_{\text{model}}}$ is a vector of the perturbation projected into the Jacobian's eigenvector space. In, 2, each projected component is squared and multiplied by the corresponding eigenvalue. Thus, if a perturbation decomposes heavily onto an eigenvector with a negative eigenvalue, our original condition for "correction" will be satisfied. $\square$

More concretely, consider the perturbation from earlier, $\mathbf{a}_t$. We can project $\mathbf{a}_t$ into eigenvector space to obtain $\mathbf{q} = \mathbf{Q}^\top\mathbf{a}_t$. Observe that entry $\mathbf{q}_i$ is the projection onto the $i$-th eigenvector with eigenvalue $\lambda_i$. From the derivation in 2, we have

$$\langle \overline{\mathbf{J}}\mathbf{a}_t, \mathbf{a}_t \rangle = (\mathbf{Q}^\top\mathbf{a}_t)^\top \mathbf{V}(\mathbf{Q}^\top\mathbf{a}_t) = \mathbf{q}^\top\mathbf{V}\mathbf{q} = \sum_i \lambda_i |\mathbf{q}_i|^2 \qquad (3)$$

Thus, if $\sum_i \lambda_i |\mathbf{q}_i|^2 < 0$, $\mathbf{a}_t$ is being corrected. Additionally, any individual $\lambda_i |\mathbf{q}|_i^2 < 0$ corresponds to an individual direction that is being corrected, specifically the $i$-th eigenvector, with correction strength $\lambda_i$. Suppose we consider the top eigenvectors, namely those with the top fifty $|\mathbf{q}_i|^2$ values; these fifty directions account for $15+\%$ of the variance of $\mathbf{a}_t$. Then, we can plot the distribution of the corresponding $\lambda_i$ to understand how TLCM interacts with the top components of $\mathbf{a}_t$.

**Results.** We compute the described plot—the histogram of $\lambda_i$ for top directions of $\mathbf{a}_t$—across 50 Jacobian-$\mathbf{a}_t$ pairs where TLCM is present and 50 pairs where TLCM is not present. Figure 4b shows the aggregated results. We observe a bimodal eigenvalue distribution for TLCM pairs. The strongest mode hovers near $\lambda = -1$, which means that a unit increase in this direction from the prior layer results in a unit-sized correction.

Importantly, a mode around $\lambda = -1$ implies correction *rather than* partial attenuation of undesired features; the features are entirely reversed. The other mode hovers around $\lambda = 0.5$, which implies that there are directions being *promoted* by TLCM, contrary to our initial expectation. This bimodal distribution is firm evidence that TLCM does not seek to attenuate the previous layer uniformly. In fact, it selects key subspaces to correct, disregarding or promoting the rest. For adjacent layers without TLCM, we observe one strong peak at $\lambda = 0$; this suggests that adjacent layers without TLCM do not interact significantly with the prior layer.

Finally, we observe that about 3000 of the total 4096 eigenvalues of $\overline{\mathbf{J}}$ are negative; see Figure 4a.

### 5.3 Propose and Reject Hypothesis

We have shown that TLCM does not uniformly reverse the prior layer's contribution but rather selectively corrects a subspace of it. To synthesize our findings, we propose the propose-and-reject hypothesis as a conceptual framework:

**Propose-and-reject hypothesis (P&R).** TLCM contributes to feature enrichment through a two-stage process: (1) a layer proposes a set of candidate features, and (2) the subsequent layer, equipped with attention mechanisms to gather context, removes irrelevant features.

P&R is consistent with our experiments thus far: Attention and MLP layers both play a role in TLCM; TLCM activates more frequently later in the context window; TLCM activates predominately in the first two-thirds of models, which is connected with enrichment; TLCM corrects *only* the prior layer; TLCM appears to perform selective correction rather than uniform attenuation, as shown in Sec 5.2; among others. We caveat that we do not explicitly test P&R in this work.

P&R could prove particularly valuable for contextual processing. For example, consider the token "202" within "1,808,202". During the forward pass, layer $i$ at this token might propose a handful of feature vectors: "hundred", "thousand", "million". Layer $i + 1$—equipped with an attention sublayer—can evaluate and eliminate the incorrect features. The incorrect features would form the "undesirable" subspace which would be corrected by layer $i+1$. The correct feature, "million", would reside in the "desirable" subspace and hence remain untouched.

## 6 Discussion

Our TLCM experiments have consequences for broader work in mechanistic interpretability. For example, since SAEs interpret residual stream contributions, TLCM's existence predicts *misfires*—features contributed to the residual stream that are immediately reversed by the subsequent layer. In addition, our research also offers a framework for explaining several challenges in SAE-based interpretability:

**Feature descriptions lack high specificity.** Recent work [47] observed that over 50% of activated features from an SAE are labeled by a large LLM as "Irrelevant" or "Only vaguely related" to the text on which they fire. Highly-activating features, though more rare, tend to show greater specificity to the text. This pattern of low specificity in most features aligns with our expectation for "misfiring" features that are subsequently corrected by the next layer.

**Effective model steering requires overcoming TLCM's correction.** While we can amplify selected features to steer model behavior, our work predicts that some amplifications will be neutralized by TLCM's corrections. As shown in Appendix E, TLCM's correction capacity begins to diminishes when the prior layer is amplified beyond $2\times$; therefore, feature amplification likely needs to exceed a critical threshold to be effective. Consistent with this hypothesis, recent work demonstrated that effective steering requires extreme amplification levels—up to $10\times$ a feature's maximum observed value [47]. We propose that solving an alternative optimization problem—intervening on feature directions that TLCM is not targeting, identified using the Jacobian—could be a promising direction for future work.

**Cross-layer transcoders outperform SAEs.** Recent work finds that the semantic meaning of the feature $\mathbf{v}_i$ and $-\mathbf{v}_i$ is unrelated [10]. For example, assume the former means "firetruck" and the latter means "Ancient Greece." If layer $i$ contributes $-\mathbf{v}_i$, an SAE is unable to determine whether layer $i$ aims to correct a faulty "firetruck" activation from the previous layer or contribute a novel "Ancient Greece" feature. Cross-layer transcoders, which are conditioned on the input residual stream, fundamentally can distinguish between these two cases, thus enabling them to learn better features—CLTs can learn both a "firetruck misfire" and an "Ancient Greece" feature that map to the same output vector. Indeed, recent work finds CLTs beat SAEs on certain metrics [40].

More broadly, we think further understanding the architectural causes for TLCM (Appendix A explores some ideas) is exciting subsequent work, as is trying to understand TLCM's corrections in feature extracted via standard interpretability techniques. We hope TLCM helps improve methods to steer LLM's forward passes and is a step towards making models more interpretable and controllable.

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

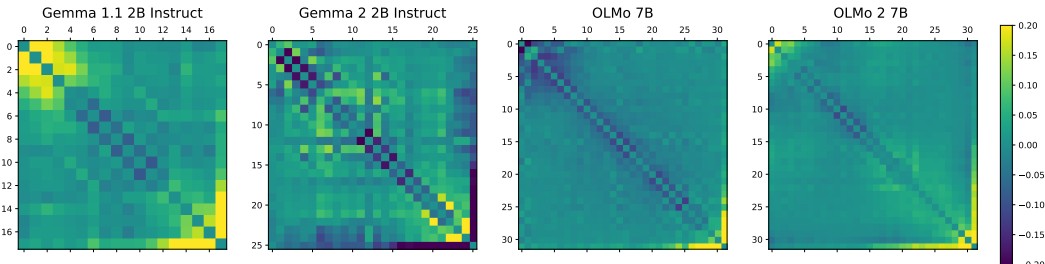

Figure A1: TLCM is consistently found on models with different approaches to LayerNorm. Surprisingly, Gemma 2 with post-LayerNorm has more pronounced TLCM. OLMo 2, which replaces pre-LayerNorm entirely exhibits TLCM. For visual clarity, we plot $\text{clamp}(\mathbf{M}, -0.2, 0.2)$ and zero the diagonals.

## A LayerNorm Blindness to explain TLCM

One natural cause of TLCM is RMSNorm. RMSNorm normalizes the input to the attention and MLP sublayers in nearly all of the open-source models analyzed. Formally, for an input $\mathbf{x} \in \mathbb{R}^{d_\mathrm{m}}$ to an attention or MLP sublayers and a learnable parameter vector $\mathbf{g}$, RMSNorm is defined as:

$$\overline{\mathbf{x}} = \frac{\mathbf{x}}{\text{RMS}(\mathbf{x})} \odot \mathbf{g} \qquad \text{RMS}(\mathbf{x}) = \frac{1}{\sqrt{d_\mathrm{m}}} \|\mathbf{x}\|_2,$$

where $d_\mathrm{m}$ is the dimension of the residual stream.

Crucially, the use of RMSNorm implies that both the attention and MLP sublayers have *layernorm blindness*; they are blind to the norm of the residual stream. This blindness is significant because these sublayers predict contributions to the residual stream (i.e. $\mathbf{x}_i + \text{Sublayer}(\mathbf{x}_i)$), whose relative impacts depend on the the residual streams current magnitude. Without visibility into the residual stream norm, attention and MLP sublayers risk under-contributing when the norm is high, which potentially leads to their contributions being overshadowed. This could encourage over-contribution behaviors followed by correction, which is consistent with TLCM.

However, we find that *LayerNorm blindness alone does not fully explain the TLCM*. This is because as discussed Sec. 5.2, the correction mechanism does not entirely reverse $\mathbf{a}_t$, contrary to what would be expected if RMSNorm were the primary cause. But perhaps more critically, models trained with alternative RMSNorm implementations continue to exhibit the mechanism:

**Gemma.** The sublayers in Gemma 1 were trained using pre-LayerNorm as described above. In contrast, Gemma 2 utilized a hybrid approach, employing both pre-LayerNorm and post-LayerNorm:

$$\mathbf{x}_{i+1} = \mathbf{x}_i + \text{LayerNorm}(\text{Sublayer}(\text{LayerNorm}(\mathbf{x}_i))).$$

The addition of post-LayerNorm should, in principle, make the residual stream norm more predictable. However, empirical results show that the correction mechanism remains robust in Gemma 2. Refer to Figure A1 for a comparison.

**OLMo.** The original OLMo models employed pre-LayerNorm exclusively. In the OLMo 2 series, pre-LayerNorm was replaced with post-LayerNorm and QK norm. Despite this architectural change, the correction mechanism persists strongly in OLMo 2, as shown in Figure A1.

## B Likelihood of Negative Cosine Similarity

### B.1 Random Normal IID Vectors

Consider two random vectors: $\mathbf{u}, \mathbf{v} \sim \mathcal{N}(\mathbf{0}, \sigma^2 \mathbf{I}_d)$.

$$\mathbb{E}[\mathrm{cossim}(\mathbf{u}, \mathbf{v})] = \mathbb{E}\left[\frac{\langle \mathbf{u}, \mathbf{v}\rangle}{\|\mathbf{u}\|_2 \|\mathbf{v}\|_2}\right]$$

$$= \mathbb{E}\left[\frac{\sum_i \mathbf{u}_i \mathbf{v}_i}{\sqrt{\sum_i \mathbf{u}_i^2}\sqrt{\sum_i \mathbf{v}_i^2}}\right]$$

$$= \mathbb{E}\left[\sum_j \frac{\mathbf{u}_j}{\sqrt{\sum_i \mathbf{u}_i^2}}\frac{\mathbf{v}_j}{\sqrt{\sum_i \mathbf{v}_i^2}}\right]$$

$$= \sum_j \mathbb{E}\left[\frac{\mathbf{u}_j}{\sqrt{\sum_i \mathbf{u}_i^2}}\right]\mathbb{E}\left[\frac{\mathbf{v}_j}{\sqrt{\sum_i \mathbf{v}_i^2}}\right]$$

$$= 0$$

The last expectations must be 0 because $\frac{\mathbf{u}_j}{\sqrt{\sum_i \mathbf{u}_i^2}}$ is distributed the same as $\frac{-\mathbf{u}_j}{\sqrt{\sum_i \mathbf{u}_i^2}}$. To calculate the variance, first observe the following fact:

$$\mathbb{E}\left[\sum_{j=1}^d \frac{\mathbf{u}_j^2}{\sum_i \mathbf{u}_i^2}\right] = 1$$

$$\sum_{j=1}^d \mathbb{E}\left[\frac{\mathbf{u}_j^2}{\sum_i \mathbf{u}_i^2}\right] = 1$$

$$d\mathbb{E}\left[\frac{\mathbf{u}_1^2}{\sum_i \mathbf{u}_i^2}\right] = 1$$

$$\mathbb{E}\left[\frac{\mathbf{u}_1^2}{\sum_i \mathbf{u}_i^2}\right] = \frac{1}{d}$$

Using this identity, the variance is:

$$\mathrm{Var}[\mathrm{cossim}(\mathbf{u}, \mathbf{v})] = \mathbb{E}[\mathrm{cossim}(\mathbf{u}, \mathbf{v})^2]$$

$$= \mathbb{E}\left[\frac{(\sum_i \mathbf{u}_i \mathbf{v}_i)(\sum_i \mathbf{u}_i \mathbf{v}_i)}{\sum_i \mathbf{u}_i^2 \sum_i \mathbf{v}_i^2}\right]$$

$$= \mathbb{E}\left[\frac{\sum_{i,j} \mathbf{u}_i \mathbf{v}_i \mathbf{u}_j \mathbf{v}_j}{\sum_i \mathbf{u}_i^2 \sum_i \mathbf{v}_i^2}\right]$$

$$= \mathbb{E}\left[\frac{\sum_i \mathbf{u}_i^2 \mathbf{v}_i^2}{\sum_i \mathbf{u}_i^2 \sum_i \mathbf{v}_i^2}\right]$$

$$= \sum_{j=1}^d \mathbb{E}\left[\frac{\mathbf{u}_j^2}{\sum_i \mathbf{u}_i^2}\right]\mathbb{E}\left[\frac{\mathbf{v}_j^2}{\sum_i \mathbf{v}_i^2}\right]$$

$$= \sum_{j=1}^d \frac{1}{d}\cdot\frac{1}{d} = \frac{1}{d}$$

## B.2 Experimental Mean and Standard Deviation

We find that the contributions of a particular layer are anisotropic; they cluster around approximately 500-800 of the 4096 dimensions of Llama's residual stream.

Specifically, we isolate the contribution vector for layer $i$ across 4096 tokens from wikitext: $\{\mathbf{c}_{i,t_1}, \mathbf{c}_{i,t_2}, \ldots, \mathbf{c}_{i,t_{4096}}\}$. After computing the SVD of each set, we plot percent variance explained by top $n$ principal components vs. $n$ in Figure A2.

Due to this anisotropy, we calculate the statistical significance of the TLCM cutoff ($-0.1$ cosine similarity) empirically. We compute the mean and variance of cosine similarity across contributions of adjacent layers on *different* tokens. More formally, we compute the mean and variance of $\text{cossim}(\mathbf{c}_{i,t_1}, \mathbf{c}_{i+1,t_2})$ for $t_1 \neq t_2$ and $4 \leq i < 20$. We find it has has mean $-0.00375$ and standard deviation is $0.03267$, meaning that our cutoff of $-0.1$ is conservatively a $3\sigma$ event; most TLCM events occur at lower cosine similarities ($-0.15$ to $-0.25$).

We plot these distributions by layer in Figure A3.

## C   Details on Jacobian Experiments

### C.0.1   Jacobian Sanity Check

We have approximated our response function $\mathbf{b}'_t$ using the Taylor expansion.

$$\mathbf{b}'_t(\boldsymbol{\Delta}) = \mathbf{b}'_t(0) + \mathbf{J}\boldsymbol{\Delta} + \mathcal{O}(\boldsymbol{\Delta}^2)$$

Here we aim to confirm that the Jacobian is appropriately representative of the transformer layer within a reasonable regime. Observe that the error of this approximation is $\mathbf{b}'_t(\boldsymbol{\Delta}) - \mathbf{b}'_t(0) - \mathbf{J}\boldsymbol{\Delta}$, and we thus denote the percent error of the approximation as follows:

$$\text{err}(\boldsymbol{\Delta}) = \frac{\|\mathbf{b}'_t(\boldsymbol{\Delta}) - \mathbf{b}'_t(0) - \mathbf{J}\boldsymbol{\Delta}\|}{\|\mathbf{b}'_t(\boldsymbol{\Delta}) - \mathbf{b}'_t(0)\|}$$

We plot the percent error error of this approximation across 46 randomly selected TLCM Jacobians for different values of $\boldsymbol{\Delta} = \alpha \mathbf{a}_t$, $\alpha \in \{-0.5, -0.4, -0.3, \ldots, 0.3, 0.4, 0.5\}$. Shown in Figure A4, we find that the Jacobian is a good approximation within this regime. For $|\alpha| < 0.1$, there is consistently around 5% error, which we believe is sufficient for our Jacobian-based analysis in Sec. 5.2.

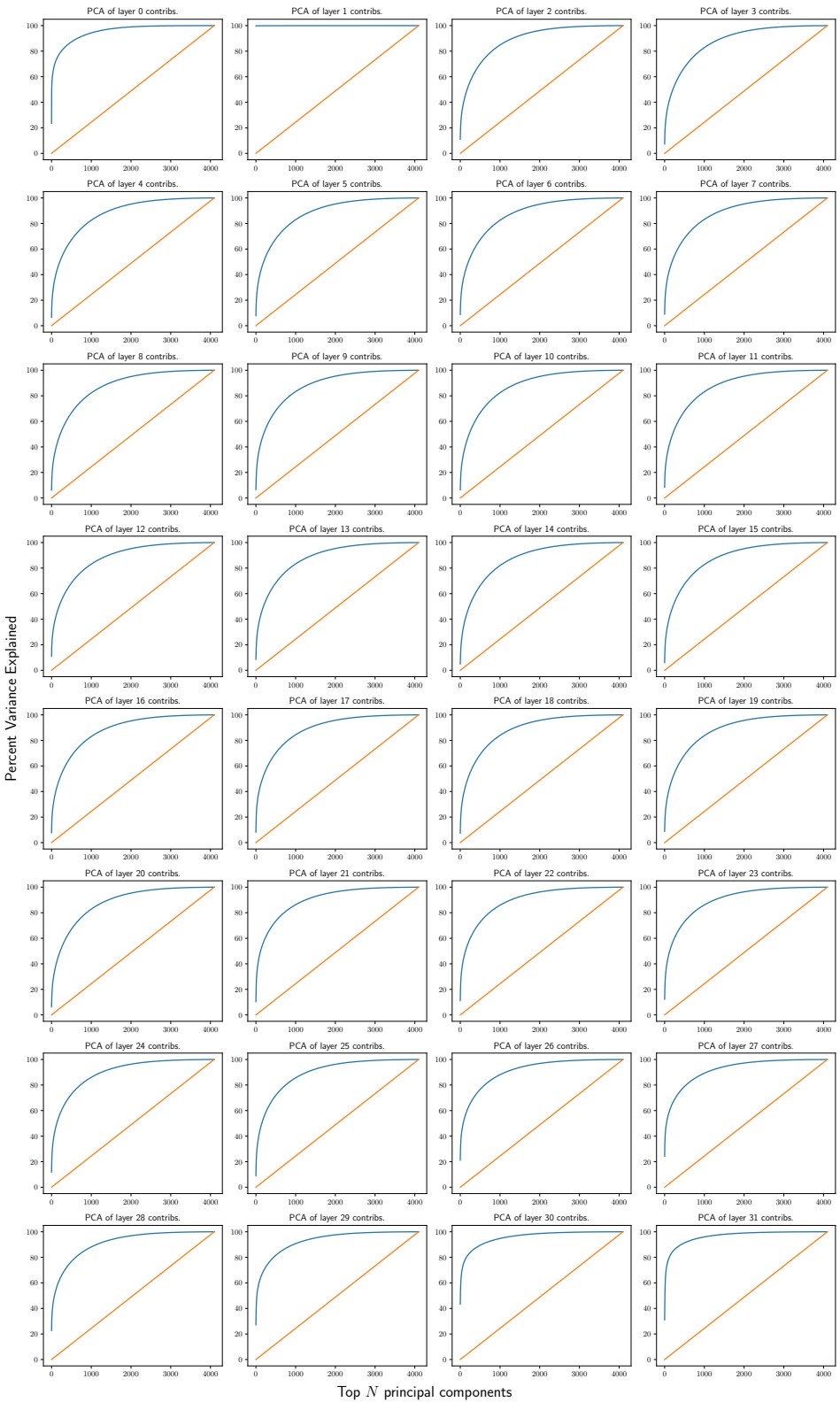

Figure A2: Percent variance explained by the top $n$ principal components vs $n$, graphed on the blue line. The orange line is what we would expect if contributions were isotropic. For layers where TLCM is most active (4 to 20), the first 500-800 principal components explain 80% of the variance in contributions, demonstrating that contributions are anisotropic.

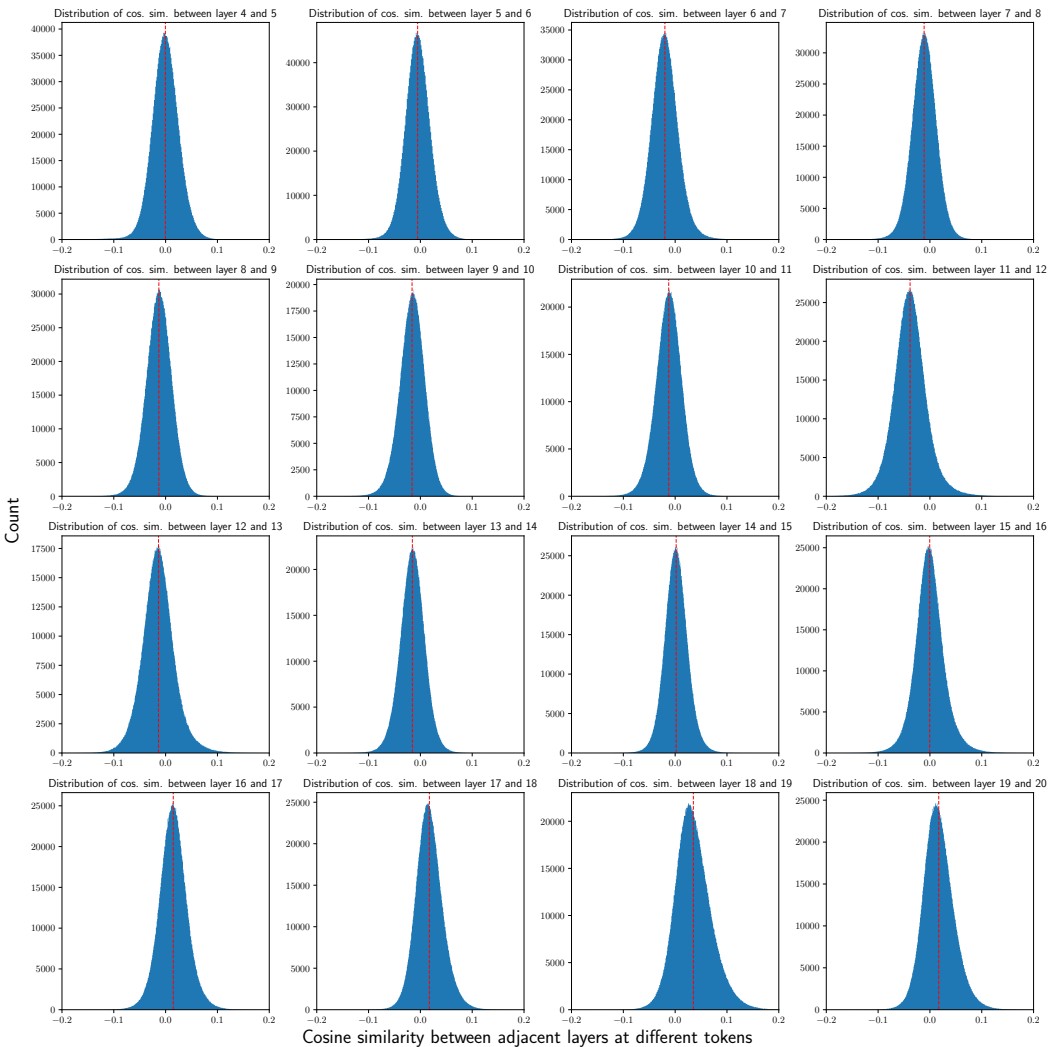

Figure A3: The empirical distribution of $\mathrm{cossim}(\mathbf{c}_{i,t_1}, \mathbf{c}_{i+1,t_2})$, $t_1 \neq t_2$ across 2000 tokens for layers $4 \leq i < 20$, corresponding to approximately 2 million samples per histogram. The red dotted line corresponds to the mean.

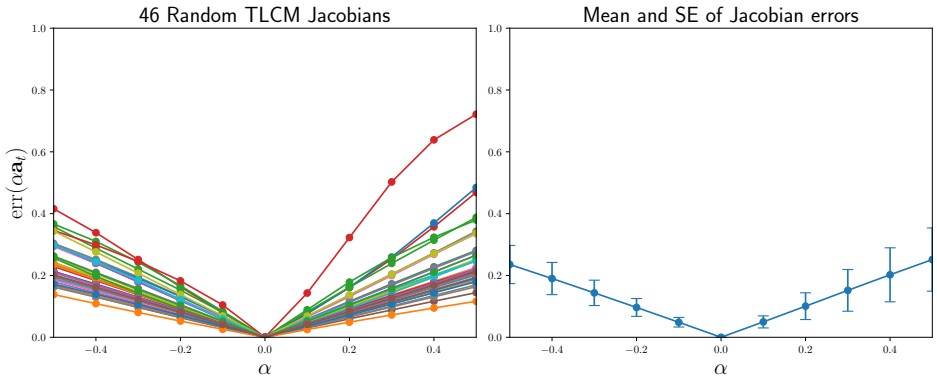

Figure A4: The transformer layer Jacobian is a very good approximation for reasonably large perturbations ($|\alpha| < 0.1$) to the following layer, making it useful for decomposing directions as described in Sec. 5.2.

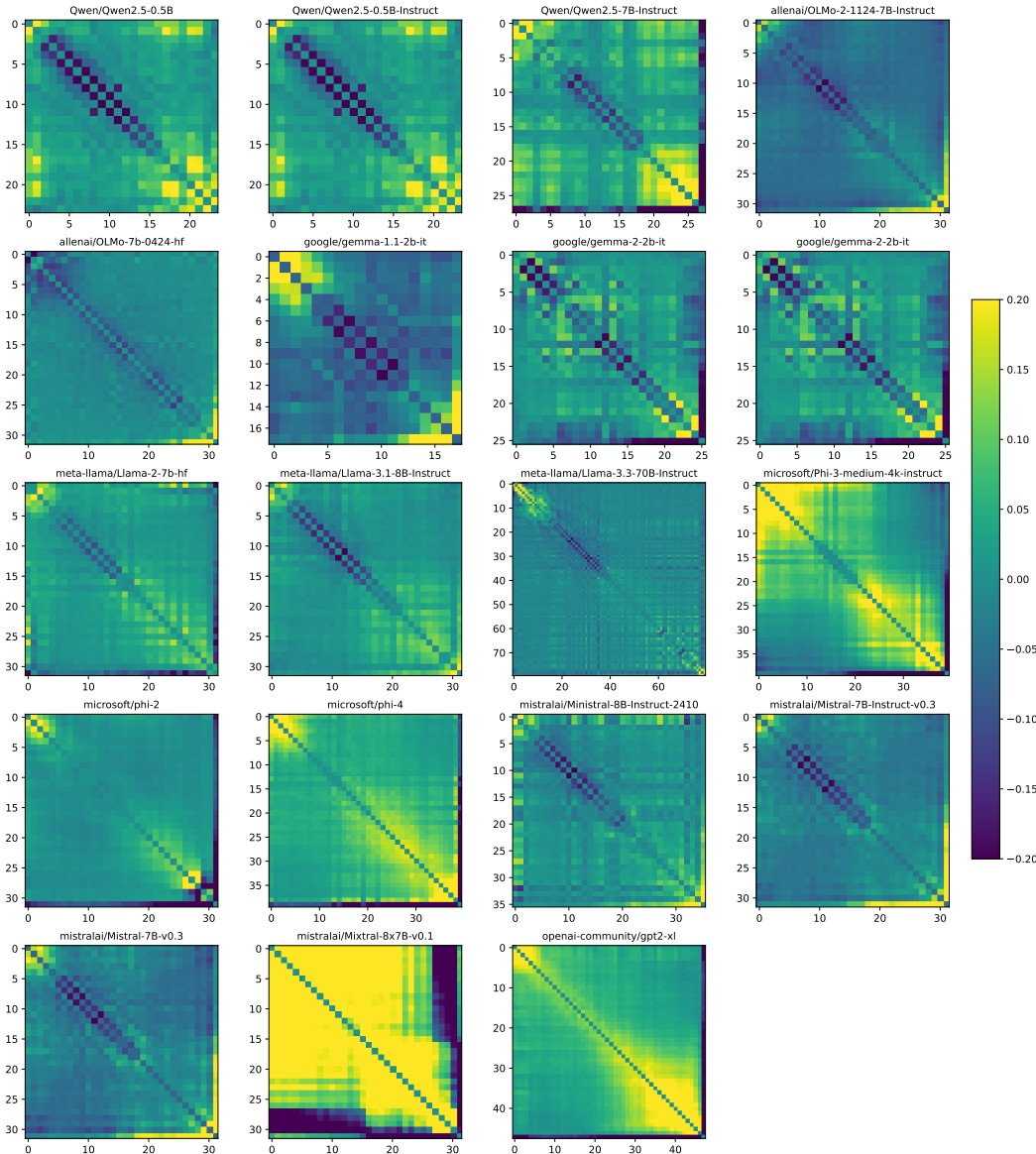

Figure A5: We plot $\text{clamp}(\mathbf{M}, -0.2, 0.2)$ for four models, zeroing diagonal entries. We computed $\mathbf{M}$ across a variety of models, pulled from Huggingface Transformers.

## D  Extended Details on TLCM Existence

In Figure A5, we plot TLCM's existence across many HuggingFace models using the same technique as described in Sec. 4.1 of the main body.

We previously demonstrated that MLPs exhibit anti-correlations with each other, while attentions do not. We additionally find that MLPs correct attentions (both within and between transformer layers) and that attentions correct MLPs from prior layers. This could be due to a common low-dimensional subspace used by both units for communication. Thus, MLPs correct both prior attention and MLPs; attentions correct just prior MLPs. Altogether, this suggests that MLPs are more responsible for TLCM's correction, but both units are involved.

Specifically, we plot $\mathbf{M}_{\text{attn}\times\text{MLP}} = \frac{1}{n}\sum_t \mathbf{M}_{\text{attn}\times\text{MLP},t}$ where
$$\mathbf{M}_{\text{attn}\times\text{MLP},t}[i,j] := \text{cossim}(\text{Attn}_i(\mathbf{x}_{i,t}), \text{MLP}_j(\mathbf{x}_{j,t}))$$
See Figure A6.

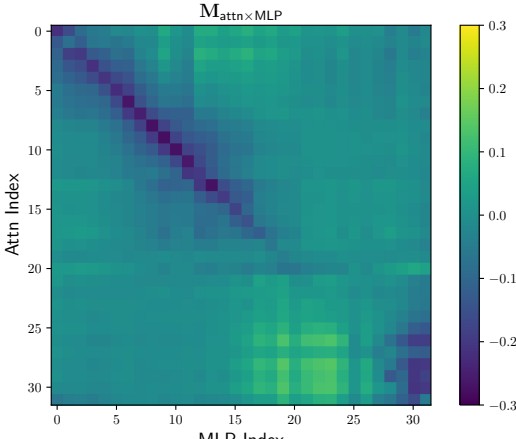

Figure A6: MLPs correct prior attentions, and attentions correct prior MLPs.

# E    Extended Details on TLCM Adaptivity

## E.1    TLCM is Adaptive Across Layers

In Figure A7, we plot TLCM's adaptive correction across different layers using the same technique as described in Sec. 4.1.

## E.2    TLCM Correction Capacity Diminishes at High $\alpha$

In Sec 5.1, we find that TLCM increases its correction as the prior layer is scaled. However, we find that as we scale the prior layer to extremely large values ($\alpha > 2$), the correction capacity of TLCM starts to diminish. Intuitively, as $\alpha$ is scaled, what TLCM previously considered a "mistake" now might be assumed "correct." This could explain why model steering interventions—in which a chosen feature vector is manually contributed to the residual stream—requires features contributions at $10\times$ the maximum ever observed value. In other words, steering interventions must overcome TLCM. See Figure A8 for plots of the correction beginning to diminish.

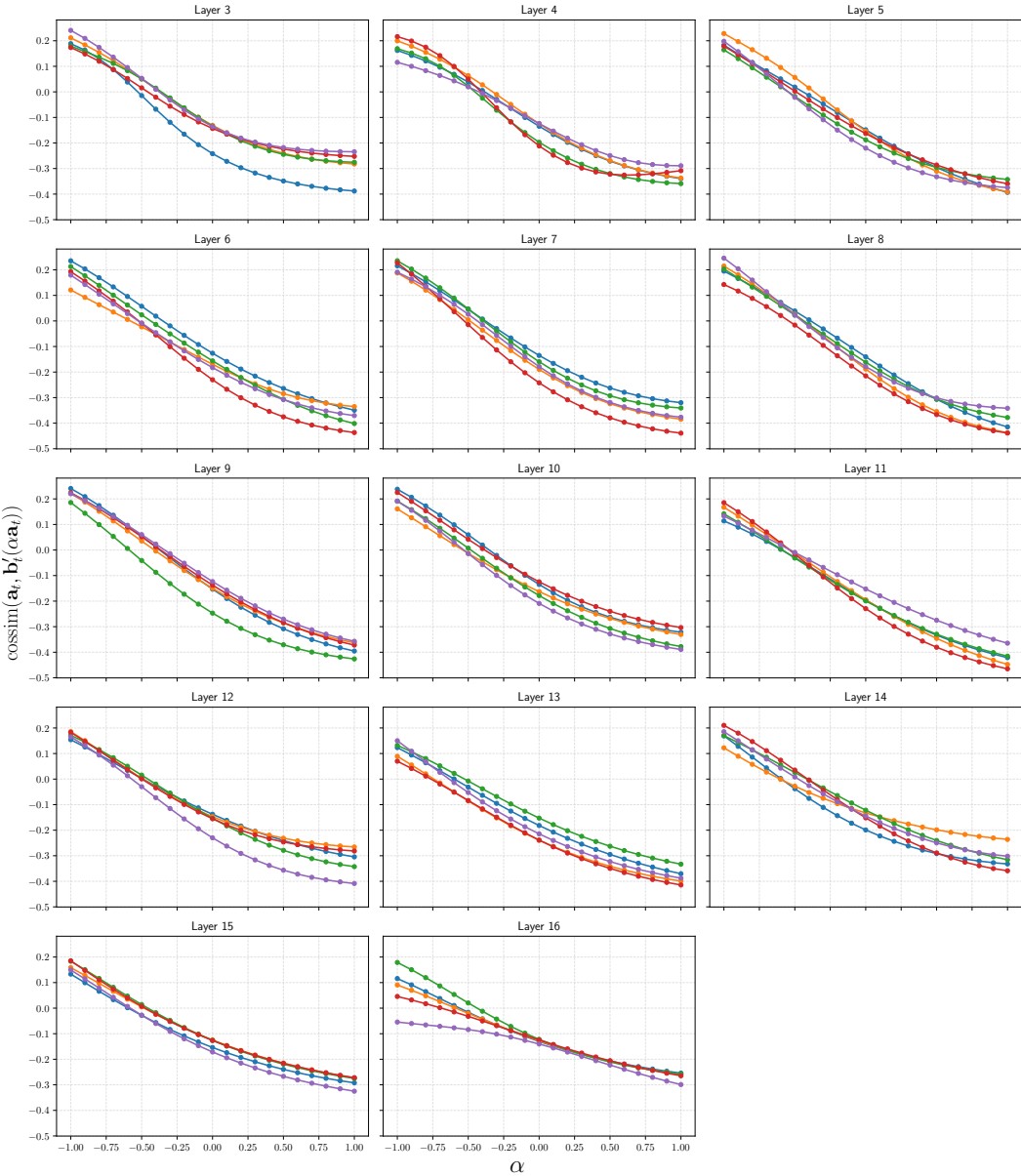

Figure A7: TLCM adaptively regulates the previous layer in a near *linear* fashion. We plot 5 random corrections across a variety of layers in Llama 3.1 8B.

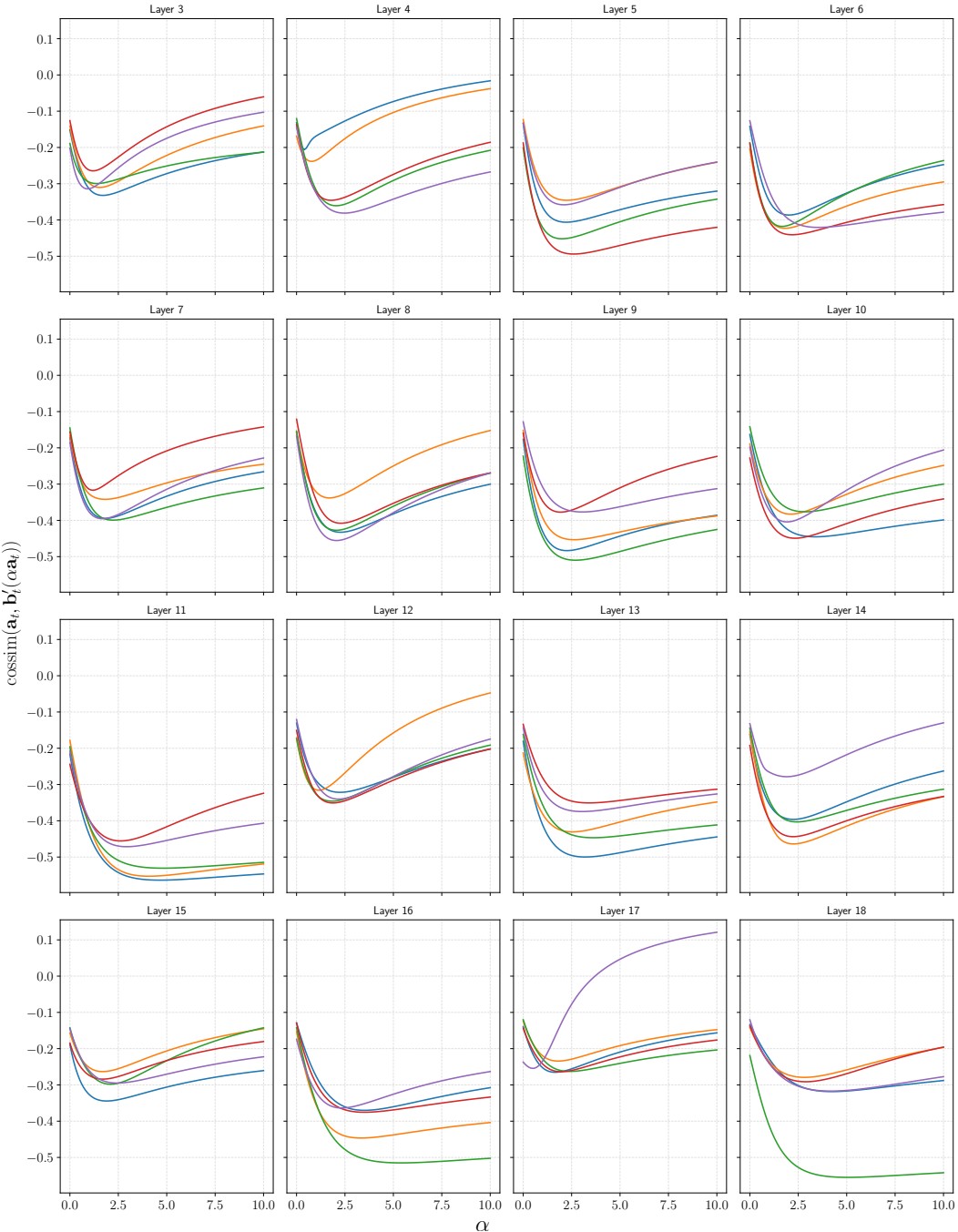

Figure A8: As we increase the previous layer more dramatically, we see TLCM's correction begin to diminish, aside from some outliers. For each layer, we sample 5 random TLCM curves and compute at $\alpha$ increments of $0.2$.

# F   OLMo Training Checkpoint Experiment Details

On a corpus of about 500 tokens of PyTorch instructional content, we compute $M$ on a handful of training checkpoints, shown in Figure A9.

The corpus is below; beyond a high enough number of tokens, we find this to have little effect on the cosine similarity matrices and thus also the figures.

```
**Using Convolutional Layers in PyTorch**
=========================================================

Convolutional layers are a fundamental component of convolutional neural
    networks (CNNs) used for image classification, object detection, and
    other computer vision tasks. In PyTorch, convolutional layers are
    implemented using the 'nn.Conv2d' module.

**Creating a Convolutional Layer**
------------------------------

To create a convolutional layer in PyTorch, you can use the following code:

'''python
import torch
import torch.nn as nn

# Define the convolutional layer
conv_layer = nn.Conv2d(in_channels, out_channels, kernel_size, stride, padding
    )
'''

*   'in_channels': The number of input channels (e.g., 3 for RGB images).
*   'out_channels': The number of output channels (e.g., 64 for a feature map).

*   'kernel_size': The size of the convolutional kernel (e.g., 3x3).
*   'stride': The stride of the convolutional kernel (e.g., 1).
*   'padding': The amount of padding to apply (e.g., 1).

**Example Usage**
-----------------

Here's an example of using a convolutional layer in a PyTorch model:

'''python
import torch
import torch.nn as nn

class ConvNet(nn.Module):
    def __init__(self):
        super(ConvNet, self).__init__()
        self.conv_layer = nn.Conv2d(3, 64, kernel_size=3, stride=1, padding=1)

    def forward(self, x):
        return torch.relu(self.conv_layer(x))

# Initialize the model and input tensor
model = ConvNet()
input_tensor = torch.randn(1, 3, 224, 224)

# Forward pass
output = model(input_tensor)
```

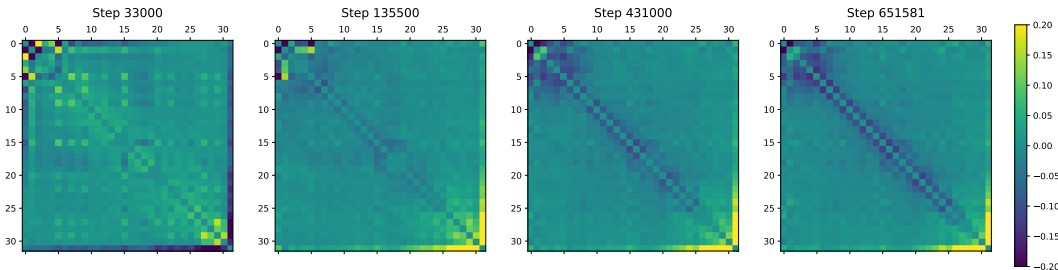

Figure A9: In OLMo 7B, TLCM emerges progressively during pretraining, with initial manifestation around step 135,500 (0.5T tokens). These four plots show $\mathbf{M}$ computed at different training checkpoints of OLMo 1B, with the rightmost plot representing the fully pretrained model. For visual clarity, we plot $\mathrm{clamp}(\mathbf{M}, -0.2, 0.2)$ and zero the diagonals.

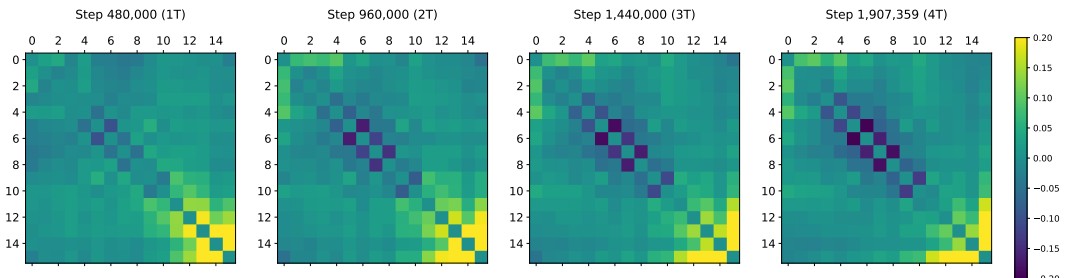

Figure A10: In OLMo 2 1B, TLCM also emerges progressively during pretraining. We plot figures at different step numbers and number of tokens (1, 2, 3, or 4 trillion tokens).

# G    Token-Level Correction Statistics

## G.1    Experiment Prompts

We use the following prompts to generate data for our experiment in Sec 4.3:

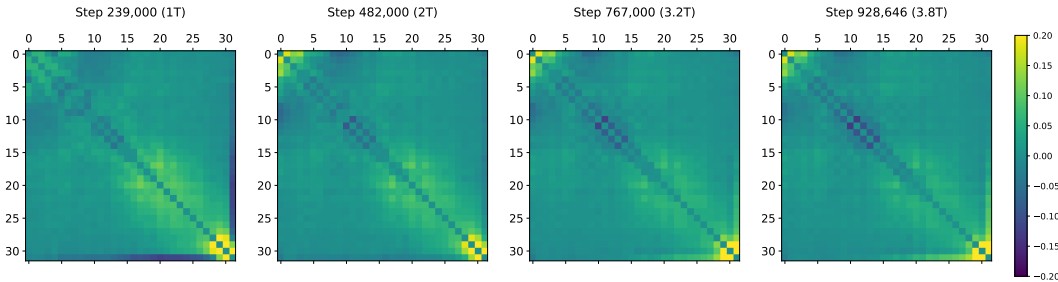

Figure A11: In OLMo 2 7B, TLCM also emerges progressively during pretraining. We plot figures at different step numbers and number of tokens.

Write a blog post about the impact of remote work on urban real estate trends.
Write an essay on the psychological effects of social media on teenagers.
Write a report detailing the advancements in renewable energy technologies over the last decade.
Write an article about the rise of plant-based diets and their environmental benefits.
Write a memo to employees explaining the new company policy on cybersecurity measures.
Write a letter to a local council advocating for improved recycling facilities in the community.
Write a proposal for implementing a mindfulness program in elementary schools to enhance student well-being.
Write a blog post about the evolution of smart home technology and its implications for privacy.
Write an essay discussing the ethical considerations of genetic editing technologies.
Write a report on the economic impacts of the COVID-19 pandemic on small businesses.
Write an article about the significance of the James Webb Space Telescope's latest findings.
Write a memo outlining the steps for a successful digital transformation in a manufacturing company.
Write a letter to a senator expressing concerns about the proposed changes to healthcare laws.
Write a proposal for a community garden project to promote local food production and community engagement.
Write a blog post about the latest trends in artificial intelligence and machine learning.
Write an essay on the role of art therapy in mental health recovery.
Write a report assessing the potential of hydrogen fuel as an alternative energy source.
Write an article highlighting the importance of biodiversity conservation in combating climate change.
Write a memo to staff regarding the integration of a new project management software.
Write a letter to an editor expressing opinions on the local government's transportation plan.
Write a proposal for a telemedicine service to increase healthcare access in rural areas.
Write a blog post discussing the future of space tourism and its possible timeline.
Write an essay exploring the cultural significance of indigenous music.
Write a report on the trends in global unemployment rates and their implications for economic policy.
Write an article about the benefits and challenges of homeschooling.
Write a memo describing the company's strategy to address the upcoming industry regulations.
Write a letter to a non-profit organization offering to partner on an environmental initiative.
Write a proposal for an employee wellness program that includes both physical and mental health activities.
Write a blog post analyzing the impact of blockchain technology on financial services.
Write an essay on the historical impact of major pandemics on societal structures.
Write a report on the viability of vertical farming in urban environments.
Write an article about the challenges of maintaining data privacy in the age of IoT.
Write a memo to update company leadership on the progress of the quarterly goals.
Write a letter to a school board proposing the introduction of coding classes in middle schools.
Write a proposal for a local government initiative to support small businesses during economic downturns.
Write a blog post about the techniques and benefits of sustainable agriculture.
Write an essay on the influence of classical music on modern genres.
Write a report on consumer behavior changes in the automotive industry towards electric vehicles.
Write an article about the role of youth activism in shaping public policy.

Write a memo detailing guidelines for handling customer data under new privacy laws.
Write a letter to the editor about the importance of public parks and open spaces.
Write a proposal for a new arts festival aiming to showcase local and international talent.
Write a blog post on the role of robotics in healthcare and potential ethical dilemmas.
Write an essay about the impact of climate change on marine ecosystems.
Write a report on strategies for managing workplace diversity in a global company.
Write an article on the resurgence of interest in vinyl records and analog music.
Write a memo to department heads about managing remote teams effectively.
Write a letter to a city planner regarding the need for improved pedestrian pathways.
Write a proposal for implementing a bike-sharing program in a mid-sized city.
Write a blog post about the future trends in education technology and their implications for learning.
Write a blog post about the growing popularity of mindfulness apps and their effectiveness.
Write an essay on the resurgence of traditional farming techniques in modern agriculture.
Write a report on the adoption of electric vehicles in major cities around the world.
Write an article about the psychological benefits of outdoor activities.
Write a memo to management detailing the steps to achieve carbon neutrality in the workplace by 2030.
Write a letter to a philanthropic organization requesting funding for a community tech hub.
Write a proposal for a series of workshops aimed at teaching digital literacy to seniors.
Write a blog post analyzing the impact of virtual reality on entertainment and media.
Write an essay discussing the philosophical implications of artificial intelligence surpassing human intelligence.
Write a report on the state of child nutrition programs in public schools.
Write an article about the role of drones in modern agriculture and their environmental impact.
Write a memo regarding the implementation of a flexible work schedule to enhance employee productivity.
Write a letter to a government official advocating for stricter air pollution regulations.
Write a proposal for a new public library with advanced digital resources.
Write a blog post about the importance of cybersecurity in the age of cloud computing.
Write an essay exploring the historical role of spices in global trade.
Write a report on the effectiveness of recent public health campaigns on smoking cessation.
Write an article on the growing trend of micro-living and tiny homes.
Write a memo introducing a new internal team dedicated to innovation and strategic initiatives.
Write a letter to parents outlining the new curriculum changes in a local school district.
Write a proposal for a mobile health clinic to serve underserved areas.
Write a blog post about the use of big data in personalized medicine.
Write an essay on the evolution of language in the digital age.
Write a report detailing the economic impact of cultural festivals on local communities.
Write an article on the significance of urban green spaces for mental health.
Write a memo to staff about upcoming training opportunities in advanced analytics.
Write a letter to the editor discussing the need for more inclusive sports programs in schools.
Write a proposal for an annual technology conference focusing on sustainability innovations.
Write a blog post about the effects of music therapy on Alzheimer's patients.
Write an essay examining the influence of video games on cognitive development.
Write a report on the future of nuclear energy and its role in combating climate change.
Write an article about the revival of handcrafts and their market in the modern economy.
Write a memo outlining the benefits of adopting a four-day workweek.
Write a letter to a university proposing a partnership for a community-based research project.
Write a proposal for developing a pedestrian-friendly zone in the downtown area.
Write a blog post on innovative approaches to waste management in urban settings.
Write an essay about the socio-economic impacts of migration on urban development.
Write a report on the adoption and regulation of cryptocurrencies in different countries.
Write an article on how to prepare pets for the arrival of a new baby.
Write a memo discussing the integration of virtual assistants into customer service.
Write a letter to a historical society proposing a project to digitize and preserve ancient manuscripts.
Write a proposal for a fitness program aimed at improving the health of office workers.
Write a blog post about the role of augmented reality in modern education.
Write an essay on the impact of global trade policies on developing economies.
Write a report analyzing the trends in youth sports and their benefits to communities.
Write an article about the ethical considerations in wildlife photography.
Write a memo to update the company on the progress of the diversity and inclusion initiative.
Write a letter to an NGO outlining a proposal for a joint clean water project in rural areas.
Write a proposal for a digital art exhibition featuring interactive installations.
Write a blog post discussing the future of autonomous public transit systems and their societal impacts.

## G.2 Correction Counts

Each tuple follows the format: (token, average TLCM activation count). We remove a few hundred tokens from the middle of this distribution due to space constraints.

Table A1: Llama 3.1 8B Instruct: Top 50 tokens with the highest average number of TLCM activations. For each token, we list the average number of times a TLCM activations occurs on the given token, aggregated across 100 long documents. Finally, we list the standard deviation and the number of occurrences of the token across the corpus to demonstrate the statistical significance.

| Token | Mean Activations | Std err. of mean | # occurrences |
|---|---|---|---|
| 202 | 16.66 | 0.16 | 265 |
| Jul | 14.00 | 0.00 | 100 |
| 26 | 13.90 | 0.04 | 112 |
| ]\n | 13.15 | 0.12 | 131 |
| Today | 13.00 | 0.00 | 100 |
| \n\n | 12.88 | 0.11 | 485 |
| ,\n\n | 12.85 | 0.16 | 26 |
| $ | 12.62 | 0.19 | 55 |
| <space> | 12.58 | 0.07 | 1276 |
| Name | 12.54 | 0.10 | 100 |
| at | 12.44 | 0.24 | 41 |
| \n | 12.41 | 0.10 | 228 |
| [ | 12.23 | 0.08 | 164 |
| assistant | 12.20 | 0.08 | 100 |
| State | 12.15 | 0.32 | 26 |
| ] | 12.08 | 0.14 | 49 |
| \t | 12.07 | 0.20 | 68 |
| Date | 12.03 | 0.24 | 30 |
| Address | 12.03 | 0.26 | 35 |
| user | 12.00 | 0.00 | 100 |
| ]\n\n | 12.00 | 0.19 | 39 |
| Date | 11.99 | 0.14 | 204 |
| 4 | 11.92 | 0.12 | 338 |
| Your | 11.81 | 0.12 | 103 |
| over | 11.74 | 0.26 | 35 |
| <space><space><space> | 11.68 | 0.19 | 57 |
| - | 11.68 | 0.13 | 112 |
| % | 11.65 | 0.14 | 52 |
| : | 11.58 | 0.08 | 606 |
| D | 11.46 | 0.23 | 28 |
| high | 11.46 | 0.23 | 39 |
| from | 11.44 | 0.14 | 109 |
| make | 11.44 | 0.20 | 50 |
| you | 11.43 | 0.15 | 96 |
| access | 11.40 | 0.15 | 89 |
| take | 11.37 | 0.21 | 30 |
| between | 11.30 | 0.26 | 27 |
| [ | 11.28 | 0.12 | 108 |
| City | 11.24 | 0.30 | 38 |
| not | 11.23 | 0.17 | 52 |
| well | 11.23 | 0.15 | 70 |
| need | 11.22 | 0.19 | 55 |
| Thank | 11.19 | 0.18 | 27 |
| 1 | 11.18 | 0.07 | 295 |
| your | 11.10 | 0.13 | 100 |
| such | 11.05 | 0.10 | 157 |
| up | 11.03 | 0.27 | 39 |
| Knowledge | 11.00 | 0.00 | 100 |
| Write | 11.00 | 0.00 | 100 |
| long | 11.00 | 0.27 | 26 |

Table A2: Llama 3.1 8B Instruct: Top 50 tokens with the lowest average number of TLCM activations. For each token, we list the average number of times a TLCM activations occurs on the given token, aggregated across 100 long documents. Finally, we list the standard deviation and the number of occurrences of the token across the corpus to demonstrate the statistical significance.

| Token | Mean Activations | Std err. of mean | # occurrences |
|---|---|---|---|
| program | 8.52 | 0.13 | 97 |
| coding | 8.50 | 0.24 | 26 |
| report | 8.49 | 0.28 | 35 |
| guidelines | 8.44 | 0.26 | 27 |
| AI | 8.42 | 0.19 | 57 |
| home | 8.41 | 0.27 | 34 |
| efficiency | 8.41 | 0.24 | 37 |
| art | 8.38 | 0.30 | 32 |
| This | 8.37 | 0.07 | 155 |
| organizations | 8.33 | 0.22 | 36 |
| community | 8.33 | 0.09 | 160 |
| ization | 8.23 | 0.27 | 39 |
| **: | 8.22 | 0.06 | 601 |
| media | 8.22 | 0.22 | 60 |
| regulations | 8.22 | 0.19 | 51 |
| -being | 8.20 | 0.20 | 59 |
| engagement | 8.17 | 0.16 | 63 |
| environmental | 8.17 | 0.20 | 42 |
| -based | 8.15 | 0.23 | 40 |
| sustainable | 8.15 | 0.11 | 89 |
| urban | 8.14 | 0.14 | 70 |
| learning | 8.12 | 0.18 | 80 |
| infrastructure | 8.12 | 0.20 | 49 |
| IoT | 8.10 | 0.25 | 31 |
| should | 8.08 | 0.14 | 59 |
| interactive | 8.08 | 0.25 | 26 |
| diversity | 8.07 | 0.24 | 28 |
| classical | 8.07 | 0.24 | 30 |
| challenges | 8.07 | 0.13 | 92 |
| cities | 8.03 | 0.18 | 40 |
| mindfulness | 8.00 | 0.29 | 33 |
| indigenous | 8.00 | 0.17 | 27 |
| agriculture | 7.96 | 0.21 | 50 |
| VR | 7.93 | 0.26 | 29 |
| workshops | 7.93 | 0.25 | 28 |
| By | 7.92 | 0.11 | 77 |
| therapy | 7.89 | 0.24 | 37 |
| trends | 7.89 | 0.27 | 27 |
| innovative | 7.89 | 0.25 | 27 |
| innovation | 7.89 | 0.18 | 35 |
| proposal | 7.86 | 0.42 | 37 |
| cognitive | 7.85 | 0.22 | 26 |
| inclusive | 7.66 | 0.17 | 41 |
| sustainability | 7.63 | 0.23 | 35 |
| tourism | 7.52 | 0.23 | 27 |
| Cities | 7.33 | 0.22 | 27 |
| ting | 7.03 | 0.05 | 102 |
| blog | 6.50 | 0.55 | 28 |
| system | 3.00 | 0.00 | 100 |
| <|begin_of_text|> | 0.00 | 0.00 | 100 |

Table A3: Gemma 2 2B Instruct: Top 50 tokens with the highest average number of TLCM activations. For each token, we list the average number of times a TLCM activations occurs on the given token, aggregated across 100 long documents. Finally, we list the standard deviation and the number of occurrences of the token across the corpus to demonstrate the statistical significance.

| Token | Mean Activations | Std err. of mean | # occurrences |
|---|---|---|---|
| `<space>` | 17.89 | 0.06 | 623 |
| 4 | 17.31 | 0.11 | 366 |
| 6 | 17.09 | 0.13 | 160 |
| `\t` | 16.56 | 0.26 | 68 |
| 2 | 16.56 | 0.06 | 1066 |
| , | 16.48 | 0.14 | 242 |
| ] | 16.39 | 0.13 | 218 |
| `<space><space>` | 16.36 | 0.09 | 657 |
| 7 | 16.17 | 0.32 | 35 |
| `<space><space><space><space>` | 16.00 | 0.27 | 57 |
| % | 15.63 | 0.29 | 57 |
| 3 | 15.61 | 0.08 | 424 |
| `\n` | 15.58 | 0.05 | 1289 |
| 5 | 15.43 | 0.12 | 183 |
| - | 15.42 | 0.08 | 840 |
| 0 | 15.32 | 0.06 | 627 |
| ], | 15.27 | 0.41 | 33 |
| Thank | 15.22 | 0.13 | 27 |
| `\n\n` | 15.16 | 0.06 | 1915 |
| $ | 15.09 | 0.25 | 55 |
| such | 15.06 | 0.13 | 157 |
| 1 | 15.02 | 0.13 | 434 |
| Today | 15.00 | 0.00 | 100 |
| 9 | 14.96 | 0.44 | 50 |
| . | 14.87 | 0.04 | 3713 |
| Address | 14.74 | 0.17 | 34 |
| recent | 14.61 | 0.29 | 33 |
| " | 14.59 | 0.30 | 49 |
| ). | 14.57 | 0.28 | 49 |
| Date | 14.49 | 0.04 | 204 |
| : | 14.49 | 0.10 | 476 |
| long | 14.46 | 0.37 | 26 |
| ) | 14.37 | 0.22 | 84 |
| them | 14.27 | 0.35 | 60 |
| Date | 14.17 | 0.18 | 30 |
| [ | 14.14 | 0.14 | 166 |
| members | 14.10 | 0.43 | 30 |
| sense | 13.97 | 0.30 | 30 |
| ( | 13.90 | 0.17 | 195 |
| modern | 13.81 | 0.32 | 32 |
| " | 13.79 | 0.23 | 43 |
| City | 13.76 | 0.30 | 38 |
| over | 13.71 | 0.27 | 34 |
| years | 13.70 | 0.35 | 40 |
| at | 13.68 | 0.35 | 41 |
| Name | 13.66 | 0.14 | 100 |
| countries | 13.59 | 0.36 | 37 |
| assistant | 13.55 | 0.14 | 100 |
| world | 13.52 | 0.28 | 54 |
| led | 13.50 | 0.28 | 44 |

Table A4: Gemma 2 2B Instruct: Top 50 tokens with the lowest average number of TLCM activations. For each token, we list the average number of times a TLCM activations occurs on the given token, aggregated across 100 long documents. Finally, we list the standard deviation and the number of occurrences of the token across the corpus to demonstrate the statistical significance.

| Token | Mean Activations | Std err. of mean | # occurrences |
|---|---|---|---|
| sustainable | 10.25 | 0.18 | 89 |
| spaces | 10.24 | 0.29 | 46 |
| plan | 10.23 | 0.39 | 39 |
| EV | 10.23 | 0.26 | 26 |
| several | 10.22 | 0.31 | 27 |
| understanding | 10.22 | 0.33 | 32 |
| growing | 10.21 | 0.24 | 38 |
| online | 10.21 | 0.17 | 43 |
| reduced | 10.19 | 0.28 | 26 |
| learning | 10.19 | 0.20 | 80 |
| environmental | 10.14 | 0.18 | 42 |
| training | 10.14 | 0.27 | 59 |
| create | 10.10 | 0.22 | 69 |
| implementing | 10.10 | 0.35 | 30 |
| mental | 10.09 | 0.21 | 53 |
| AI | 10.09 | 0.21 | 57 |
| indigenous | 10.07 | 0.29 | 27 |
| local | 10.07 | 0.16 | 110 |
| significant | 10.06 | 0.16 | 143 |
| comprehensive | 10.05 | 0.22 | 39 |
| together | 10.04 | 0.41 | 26 |
| blog | 10.00 | 0.07 | 28 |
| address | 10.00 | 0.33 | 41 |
| innovative | 9.96 | 0.29 | 27 |
| promoting | 9.94 | 0.24 | 47 |
| post | 9.94 | 0.22 | 31 |
| develop | 9.89 | 0.25 | 35 |
| benefits | 9.87 | 0.19 | 105 |
| VR | 9.86 | 0.28 | 29 |
| community | 9.83 | 0.17 | 160 |
| awareness | 9.80 | 0.41 | 30 |
| prioritize | 9.79 | 0.29 | 28 |
| promote | 9.74 | 0.15 | 102 |
| progress | 9.72 | 0.38 | 29 |
| following | 9.70 | 0.31 | 27 |
| clear | 9.69 | 0.33 | 26 |
| concerns | 9.67 | 0.32 | 48 |
| approach | 9.65 | 0.34 | 34 |
| complex | 9.61 | 0.27 | 38 |
| By | 9.58 | 0.17 | 77 |
| challenges | 9.53 | 0.25 | 92 |
| improved | 9.52 | 0.28 | 33 |
| interactive | 9.42 | 0.38 | 26 |
| enhance | 8.97 | 0.34 | 35 |
| improve | 8.91 | 0.20 | 70 |
| explore | 8.74 | 0.29 | 39 |
| feedback | 8.70 | 0.28 | 27 |
| mitigate | 7.88 | 0.37 | 26 |
| Write | 7.00 | 0.00 | 100 |
| <bos> | 1.00 | 0.00 | 100 |

