# OpenReview forum: "LLM Layers Immediately Correct Each Other"
_NeurIPS.cc/2025/Conference — NeurIPS 2025 poster_

### Official Review · Reviewer_VpPo · 2025-06-02

**Clarity:** 3
**Significance:** 4
**Originality:** 3
**Rating:** 5
**Confidence:** 4

**Summary:**

The paper introduces the observation termed Transformer Layer Correction Mechanism (TLCM): Adjacent layers in many Transformer LLMs seem to partially counteract each other’s contributions (negative cosine similarity). The paper provides convincing evidence of the existence of this effect (particularly in Figures 1 and 3), and shows that it originates from the feed-forward layers. The paper furthermore proposes a “propose-and-reject” hypothesis as an explanation for the TLCM, and attempts to use this to explain challenges in SAE-interpretability.

**Questions:**

I have the following questions for the authors:

1) Did you subtract the dataset mean before computing the cosine similarity? I wonder if a non-zero mean could produce an offset in the cosine similarity.

2) In Appendix A3 you provide cosine similarity histograms for the t1 != t2 case split up by layer. I would like to see the t1 = t2 histogram (Figure 2b) split up by layer as well, maybe in an Appendix. It would be interesting to know whether the TLCM mechanism is bimodal in individual layers.

3) In line 273 you say that “The mechanisms that handle the latter two tasks must be non-trivial as the previous layer’s contribution is hidden inside the residual stream soup” – do the superposition hypothesis, and the observation that vectors in high-dimensional spaces are often nearly-orthogonal, affect the difficulty of this?

4) In the propose & reject hypothesis you propose that the following layer uses context to correct the predictions of the previous layer. This makes sense in the first (few) layers, but by the midpoint of the model the amount of context available to different layers (number of preceding attention layers) seems quite close. Do you still think this is the reason for the TLCM?

5) In section 5.2 you ask “is the entire layer scaled back uniformly or are certain subspaces targeted for correction?” – scaling the entire layer back seems like a simplistic alternative, of course that wouldn’t be happening. Are there other alternative hypotheses?

6) Line 293 introduces the Jacobian J – what are the dimensions / indices of this Jacobian? Is it the embedding dimension? How are derivatives across-token positions handled?

7) In lines 334 to 355 you discuss how TLCM could cause certain SAE failure modes. This part would be more convincing if you included experiments e.g. checking whether low-specificity SAE features line up with negative-eigenvalue-subspaces.

**Ethical Concerns:**

["NO or VERY MINOR ethics concerns only"]

**Final Justification:**

What issues were resolved, which issues remain unresolved:

> The proposed “propose-and-reject” hypothesis is only weakly supported – it is a very specific mechanism, and I expect that the experiments would equally support alternative hypotheses. Additionally, as the paper mentions, it is unclear whether the full range of results in Figure 3 can be explained by the hypothesis, or whether an unrelated self-repair mechanism is involved [lines 268-269]. The paper would be improved by considering more alternative explanations for the observations.

I appreciate the clarification of the propose-and-reject hypothesis and its context in the rebuttal.

> The eigenvalue analysis of the Jacobian describes the effect of layers in terms of subspaces. It is unclear how this explanation fits together with the superposition hypothesis (which the authors seem to believe, based on the equation above line 90).

I believe this question hasn't been clearly addressed, but I'm not confident this is a big issue.

---

I did not find the terminology or lack of rigor to be an issue when reading the paper.

---

Overall this paper is clearly one of the most interesting recent works in mechanistic interpretability, and I maintain my rating of 5: Accept. I increased my Quality score from 3 to 4 upon reviewing the rebuttal.

**Limitations:**

The authors discuss some limitations throughout the paper. While a dedicated paragraph/section to discuss limitations would be easier to understand, it may not be strictly necessary.

**Paper Formatting Concerns:**

No major issues.

**Quality:**

4

**Strengths And Weaknesses:**

**Strengths:**

The TLCM mechanism observed in this paper is, as far as I know, novel and different to previous studies focused on attention heads [30, 31]. The results are striking and promise insight into the internal mechanisms of transformers.

The cosine similarity measurements over layers (Fig 1, 2a) and context length (Fig 2b) are clear experiments.

The comparison with a baseline of cosine similarity (lines 131 through 142) are great and help establish confidence in the statistical significance of the result. (Though it is not clear whether the “different tokens” baseline is the right comparison to compute the 3 sigma number.)

The intervention experiments (Figure 3) are impressive, showing an almost linear response of the correction layer to amplifying the previous layer contribution.

**Weaknesses:**

The proposed “propose-and-reject” hypothesis is only weakly supported – it is a very specific mechanism, and I expect that the experiments would equally support alternative hypotheses. Additionally, as the paper mentions, it is unclear whether the full range of results in Figure 3 can be explained by the hypothesis, or whether an unrelated self-repair mechanism is involved [lines 268-269]. The paper would be improved by considering more alternative explanations for the observations.

The eigenvalue analysis of the Jacobian describes the effect of layers in terms of subspaces. It is unclear how this explanation fits together with the superposition hypothesis (which the authors seem to believe, based on the equation above line 90).

In lines 334 to 355 the paper discusses TLCM explaining issues with SAEs and steering. I was not convinced by this that TLCM would explain the SAE and steering issues. (see “Questions” for suggestions)

Section 5.2 is less clear than the rest of the paper, specifically how the Jacobian is computed is unclear (see below), and the proof is hard to follow.

*Formatting issue: Equation labels are partially missing; equation (3) is referenced but not labelled. It would be good for the early equations in the paper to have labels.*

*Minor typo: “begins to diminishes”*

---

> ### Author Rebuttal · Authors · 2025-07-31
>
> We’re glad you enjoyed our work, finding our results “striking,” “novel and different from previous studies,” and “impressive\!” We answer your questions below:
>
> >Did you subtract the dataset mean before computing the cosine similarity? I wonder if a non-zero mean could produce an offset in the cosine similarity.
>
> Good question. We did not subtract the mean. However, we think:
>
> * The distribution of cosine similarities (see: Figure 2b) is ***strongly*** bimodal, which we think is the strongest evidence that there are (at least) two modes of layer operation: TLCM and non-TLCM.
> * We think it’s more interesting & surprising that the absolute cosine similarity sits strongly below zero, rather than the relative/z-scored cosine similarity is low. This is contrary to our expectation that contributions between layers should have positive cosine similarity (or at minimum be orthogonal), as we discuss in L. 99-105.
>
> >In Appendix A3 you provide cosine similarity histograms for the t1 \!= t2 case split up by layer. I would like to see the t1 \= t2 histogram (Figure 2b) split up by layer as well, maybe in an Appendix. It would be interesting to know whether the TLCM mechanism is bimodal in individual layers.
>
> Thanks for raising this\! We included this exact plot in Figure 2b (observe the strong bimodality) [L. 127-130]. We realize this should be made more prominent, so we’ve updated Sec. 4.1 to discuss this plot more.
>
> >In line 273 you say that “The mechanisms that handle the latter two tasks must be non-trivial as the previous layer’s contribution is hidden inside the residual stream soup” – do the superposition hypothesis, and the observation that vectors in high-dimensional spaces are often nearly-orthogonal, affect the difficulty of this?
>
> Interesting point. Yes. The superposition hypothesis suggests that the residual stream will be some linear combination of feature vectors. We think identifying and correcting the features specifically contributed by the previous layer is a difficult task, unless there is some commonality between those features. For example, the features are wrong. This directly leads us to the *propose-and-reject* hypothesis.
>
> >The proposed “propose-and-reject” hypothesis is only weakly supported
>
> The propose-and-reject hypothesis states that a layer proposes a broad swath of possible “features,” while the following layer—equipped with an attention head to gather context—removes the features that are not relevant.
>
> We offer the propose-and-reject hypothesis as a conceptual synthesis that is consistent with all our empirical results thus far. Specifically, the hypothesis explains:
>
> * Quantitative contextual dependency experiments.
> * TLCM corrects *only* the most recent layer, rather than correcting *all prior layers*
> * TLCM’s prevalence in early layers, which are linked to “enrichment”
> * Very low TLCM activations on beginning of sequence tokens
> * Attention/MLP experiments which show that TLCM emerges when considering ***their combination*** (MLPs cannot access context without attention)
> * The causal relationship we show—where the following layer linearly & dynamically attenuates the prior layer—in Sec 5.2.
> * Sec 5.3’s Jacobian experiment demonstrating specific reinforcement and **unit** attenuation ($\\lambda \= 1$) of features
>
> With this framing, we intend to **generate future research directions**, not to close the discussion.
>
> >In the propose & reject hypothesis you propose that the following layer uses context to correct the predictions of the previous layer. This makes sense in the first (few) layers, but by the midpoint of the model the amount of context available to different layers (number of preceding attention layers) seems quite close. Do you still think this is the reason for the TLCM?
>
> Good point. This seems reasonable and likely explains why the prominence of TLCM **decreases in the final two-thirds of model layers**. Figure A5 illustrates this phenomenon across many models.
>
> >In section 5.2 you ask “is the entire layer scaled back uniformly or are certain subspaces targeted for correction?” – scaling the entire layer back seems like a simplistic alternative, of course that wouldn’t be happening. Are there other alternative hypotheses?
>
> In Appendix A, we provide a theoretical rationale for why the model might fully “scale back” the previous layer. Essentially, we introduce the concept of LayerNorm blindness, which is the phenomenon that LayerNorm causes attention and MLP sublayers to be “blind” to the norm of the residual stream. Thus, they do not know how to size their contributions and may be biased to over-contribute so their contribution is not overshadowed. However, we find TLCM in models that do not have LayerNorm blindness, contradicting this explanation.
>
> This fact (along with our Sec 5.2 Jacobian eigenbasis and Sec 4.3 experiments) strongly suggest that TLCM is responsible for attenuating a small subspace of the prior layer, leading us to the *propose-and-reject* hypothesis, in which layers are correcting incorrect features added by the prior layer.
>
> >Line 293 introduces the Jacobian J – what are the dimensions / indices of this Jacobian? Is it the embedding dimension? How are derivatives across-token positions handled?
>
> We take the layer Jacobian $\\mathbf{J} \\in \\mathbb{R}^{d\_\\text{model} \\times d\_\\text{model}}$, which includes both the attention and MLP sublayers. We are taking the Jacobian with respect to the output of the prior layer, so the effects of other tokens are considered (linearly) in this Jacobian. In practice, this is materialized by computing the gradient of the layer's 4096 outputs, one at a time using PyTorch.
>
> >Section 5.2 is less clear than the rest of the paper, specifically how the Jacobian is computed is unclear (see below), and the proof is hard to follow.
>
> Great point! We have moved the bulk of the proof's content to the appendix and increased its clarity. In its place, we've included details on how we compute the Jacobian and the intution for the eigenbasis approach.
>
> >In lines 334 to 355 the paper discusses TLCM explaining issues with SAEs and steering. I was not convinced by this that TLCM would explain the SAE and steering issues. (see “Questions” for suggestions)
>
> Thanks for giving this thought. We're curious to understand your concerns with this explanation.
>
> >In lines 334 to 355 you discuss how TLCM could cause certain SAE failure modes. This part would be more convincing if you included experiments e.g. checking whether low-specificity SAE features line up with negative-eigenvalue-subspaces.
>
> Good suggestion; we agree this would be valuable. Anthropic has predominantly completed the most substantial work with SAEs and high-quality labeled features. However, they release neither their model weights or learned features. We are excited about open-source SAEs with great natural language annotations, which would make experiments like these possible. For now, we see it as important future work.

---

> > ### Comment · Reviewer_VpPo · 2025-08-04
> >
> > >> I would like to see the t1 = t2 histogram (Figure 2b) split up by layer as well, maybe in an Appendix. It would be interesting to know whether the TLCM mechanism is bimodal in individual layers.
> >
> > > Thanks for raising this! We included this exact plot in Figure 2b (observe the strong bimodality) [L. 127-130]. We realize this should be made more prominent, so we’ve updated Sec. 4.1 to discuss this plot more.
> >
> > But Figure 2b doesn't show the histogram split up by layer, does it?
> >
> > ---
> >
> > I want to thank the authors for their detailed rebuttal. This paper looks like one of the most exciting recent works in mechanistic interpretability. While I can't increase my rating to 6 ("flawless", "groundbreaking") I maintain my accept recommendation.
> >
> > I further would like to note that I did not find the terminology or lack of rigor to be an issue when reading the paper.

---

### Official Review · Reviewer_dNpi · 2025-06-29

**Clarity:** 2
**Significance:** 3
**Originality:** 3
**Rating:** 5
**Confidence:** 4

**Summary:**

The paper documents a novel phenomenon of 'Transformer Layer Correction Mechanism'; adjacent layers in transformers sometimes have 'anti-correlated' activations. They observe that this phenomenon occurs across number of open-source LLMs, though is absent in models that use dropout (Phi 4, GPT2-XL).

**Questions:**

1. Line 162-163: "Appendix Figure A4 demonstrates that the mechanism’s strength increases markedly during the latter two-thirds of training, suggesting a systematic development pattern." I could not understand figure A4 from its caption. My best guess is that authors meant some other figure here (probably A9)? Or otherwise figure A4 needs to be explained better.

2. Figure 2a is referenced much after 2b. Flip the order of the figure.

4.Line 210-211. What does a TLCM 'activation' mean? This concept is not defined anywhere (or may be I missed it? In which case, it ought to be defined more saliently). I in general was not able to understand the y-axis of figure 2b(left).

5. Why does Qwen 0.5B has this phenomenon but not GPT-2 XL? Is dropout being present/absent the sole determinent of this phenomenon?

6. Figure 3 -- why is this done over 5 random tokens only? Can't it be done at a larger scale?

6. Compen

**Ethical Concerns:**

["NO or VERY MINOR ethics concerns only"]

**Final Justification:**

I am keeping my original score. See the strengths and weaknesses as listed above for why so.

**Limitations:**

Limitations are not explicitly discussed. Authors should address limitations and open questions comprehensively to invite further research on this topic.

**Paper Formatting Concerns:**

None.

**Quality:**

3

**Strengths And Weaknesses:**

### Strengths
- The phenomenon documented in the paper is novel and interesting.
- The paper is mostly easy to follow.
- The paper mosty does a good job of conducting systematic studies.


### Weaknesses
1. Figure captions can be improved in many cases.
2. Some results (e.g., Figure 3) are over very small sample sizes.
3. Compensatory effect (line 263-269) is not sufficiently explored. I think this effect goes against 'propose and select' hypothesis so would be worth exploring further, and understanding. It is also not clear why it has to be adjacent layers that do the 'propose and select' operation.
4. There should be more commentary on why some models demonstrate TLCM but not others.

### Improvements
I think doing the following would help improve the paper (though I don't think they are 'weaknesses'):

1. Including analysis of some state space models (e.g., Mamba) and seeing whether this phenomenon is particular to Transformers architecture or occurs across autoregressive LLMs.
2. Including analysis of models of some modality, e.g., vision.
3. Compensatory effect (line 263-269) is worth exploring more.
4. It would be interesting to develop some sort of improvement to intervention vectors based on TLCM. Naively seems like adding intervention vectors at alternate layers might be the best strategy to overcome TLCM?

---

> ### Author Rebuttal · Authors · 2025-07-31
>
> We’re glad you thought our work was “novel and interesting” and conducts good “systematic studies\!” Thanks for your questions; we address them below:
>
> >Figure 3 \-- why is this done over 5 random tokens only? Can't it be done at a larger scale?
>
> Thanks for raising this. We include full results for this experiment (total of 135 randomly sampled tokens across 18 layers) in Appendix E, Figures A7 and A8. We see that the results in Figure 3 are largely representative of the appendix figures.
>
> >Line 210-211. What does a TLCM 'activation' mean? This concept is not defined anywhere (or may be I missed it? In which case, it ought to be defined more saliently). I in general was not able to understand the y-axis of figure 2b(left).
>
> Great question. We define a TLCM activation as any pair of adjacent layers whose contributions’ cosine similarity is less than \-0.1. More formally, $\\operatorname{cossim}(\\mathbf{c}\_{i,t}, \\mathbf{c}\_{j,t}) \< \-0.1$ (L. 174-175). Note that an LLM with $n$ transformer layers can have up to $n-1$ activations. The y-axis of Figure 2b (left) is the average number of TLCM activations on the $i$th token across a corpus of 2,000 wikitext documents.
>
> We’ve updated our wording in this section (and Sec 4.3) to explicitly define “TLCM activation,” to your suggestion.
>
> >Why does Qwen 0.5B has this phenomenon but not GPT-2 XL? Is dropout being present/absent the sole determinent of this phenomenon?
>
> This remains an open question.
>
> We have not seen TLCM on any model with dropout, which indicates that dropout could be involved. However, GPT-2 is also undertrained compared to modern models, being trained on at most 100B tokens. Observe in Figure A9 that the OLMo 1B models (similar size to GPT-2 XL) do not exhibit any traces of TLCM until being trained on 500B tokens. We’ve run a similar test on OLMo-2 7B and observe that TLCM starts to appear around 2T tokens. For additional context, Llama-3 8B was trained on 15T tokens\!
>
> >It would be interesting to develop some sort of improvement to intervention vectors based on TLCM. Naively seems like adding intervention vectors at alternate layers might be the best strategy to overcome TLCM?
>
> We agree this is an important area of future work. That’s an interesting approach: it seems like even if you add an intervention vector at layer $i$, our results indicate that you will get a correction at layer $i+1$. Although it’s possible if you add interventions at both layers, this correction breaks down. We’re interested in seeing how this plays out.
>
> >Line 162-163: "Appendix Figure A4 demonstrates that the mechanism’s strength increases markedly during the latter two-thirds of training, suggesting a systematic development pattern." I could not understand figure A4 from its caption. My best guess is that authors meant some other figure here (probably A9)? Or otherwise figure A4 needs to be explained better.
>
> Thank you for catching this. You are right–we are referring to Figure A9. Because the paper and appendix were submitted separately at different times, it seems like we unintentionally compromised the correspondence of this figure in the paper and in the appendix. This is fixed for the camera-ready version.
>
> >Figure 2a is referenced much after 2b. Flip the order of the figure.
>
> Good point; we’ve fixed this.
>
>
> We’re happy to address any other questions you have.

---

> > ### Comment · Reviewer_dNpi · 2025-08-05
> >
> > Thanks for the response, and answering my questions. I don't have any other questions.

---

### Official Review · Reviewer_ucnA · 2025-06-30

**Clarity:** 1
**Significance:** 2
**Originality:** 2
**Rating:** 2
**Confidence:** 3

**Summary:**

This paper discusses TLCM; a mechanism empirically observed in transformer architectures, where the “contribution” vector, i.e., the difference between intermediate representations, exert negative similarity with the contribution vectors of immediately adjacent layers (while with later layers, the similarity becomes positive again). Through a cosine similarity analysis, they initially show that this effect seems to occur on the majority of the evaluated model families (LLaMa, Qwen, Mistral, …) and that it mainly appears in early layers.  The authors claim this is unexpected, as this seems to show that transformer layers systematically reverse the contributions of prior layers. Through evaluation studies, the authors further evaluate when this effect emerges and whether it occurs consistently across inputs. They find that (1) TLCM emerges during initial phases of pretraining, and (2) list some tokens which seem to be more and less susceptible to “correction”. Finally, the authors construct the “propose-and-reject” hypothesis where one layer proposes candidate features, and the next layer, using additional context, selectively rejects or reinforces them. To formalize this, the authors analyze the Jacobian of the next layer and claim that eigenvectors with negative eigenvalues correspond to directions that are being corrected. Empirical results suggest that correction does not apply uniformly across the residual stream but targets specific subspaces.

**Questions:**

- L.40 How do we quantify “high context dependency”? What makes a date, number, or punctuation tokens more “context dependent” than others?
- L.99-101: how do we know that contribution vector similarity tells us something about “novel information” and “reinforcing existing representations”?

**Ethical Concerns:**

["NO or VERY MINOR ethics concerns only"]

**Final Justification:**

As is visible in my review as well as in follow-up discussion, in my opinion, this work raises an interesting questions coupled with interesting empirical observations, however, the paper lacks (a) a solid theoretical foundation and (b) empirical rigor to convincingly and quantifiably prove the generalizability of the found phenomena. The paper aims to characterize TLCM from too many angles, without going into too much detail for any of them. Thus, many of the made claims are speculative and are, thus, overstated in the paper, in my opinion. Follow-up discussions with the authors helped clear up some confusions, however, they failed to make the empirical results more convincing.

**Limitations:**

- the generalization of the results is a significant limitation, as mentioned above.

**Quality:**

1

**Strengths And Weaknesses:**

### Strengths

- The paper makes empirical contributions about the representational similarity in adjacent transformer layer representations, which re-appear across multiple architectures. This is interesting. They also aim to understand the problem from different perspectives (e.g., effects of training, token classes, …).
---
### Weaknesses

Terminology is loosely used and many claims are vague or unfounded. A few examples:
 - E.g., L.29-37; what does it mean to “partially oppose” in a general setting, or “reverse” other layers’ contributions? Similarly, L.52 “verify the features and selectively correct inappropriate ones”; these should be more precise, in its core the paper observes representational contribution vector similarity
- L.40 How do we quantitatively measure “high context dependency” in tokens? It seems to me that the definition is more speculative than quantitative.
- L.99-101: how do we know that contribution vector similarity tells us something about “novel information” and “reinforcing existing representations”?
- L.143: Having negative cosine similarity does not mean “erasing information”

Although the paper aims to evaluate the phenomenon of TLCM expansively, a lack of empirical rigor and ablation makes the results and conclusions seem speculative, rather than generalizable. A few examples:

- L.115-117: the authors compute the cosine similarity between contribution vectors, but report on “correlation” in L. 116 and throughout.
- L. 128: lack of more rigorous derivation/description why choosing exactly $-0.1$ for the threshold for TLCM. “Seems to have high specificity” (L.130) does not make it well-defined. Looking at Figure 2b) this is not even at the low point of the $\text{cossim}$ distribution.
- §4.2 raises three interesting potential scenarios about when TLCM may arise, but they only evaluate a single one of them on a single architecture (OLMo 1B).
- The authors show that TLCM appears across various architectures, yet individual studies §4.2 and §4.3 are only done on a single model, making it difficult to understand the generalizability of the findings.
- §4.3 only reports qualitative results, where, according to the description, quantitative evaluation was performed.
- Figure 3 only shows layer 8 for an unknown model, and 5 “randomly sampled tokens”.
- The derivation of the “propose-and-reject” hypothesis seems rather speculative.

Lack of formal rigor: many of the findings are purely empirical, however, there lacks a convincing, rigorous theoretical discussion about TLCM, or why we are surprised by the empirical findings (elaborations of L. 99-105). This would complement the empirical results well.

Significant writing issues make the manuscript hard to follow and the contributions hard to identify.
- E.g., the introduction is written vaguely without explaining the approach or contribution (e.g., L.55-58), discussion sections heavily overlap with the main body
- Additionally there are quite a few typos and notational problems (e.g., §3, or see typos)

---
### Typos:
- L.67-69: citations
- L. 71 “large language models” -> “LLMs”
- L. 76 “onn” -> “on”
- L. 84 “unembedded”?
- Restating an introduced quantity, L.159
- L. 175: “this corpora” -> “this corpus”
- L.259: “numners” -> “numbers”
- ...

---

> ### Author Rebuttal · Authors · 2025-07-31
>
> Thank you for your thoughtful feedback\! We respond to your questions below.
>
> >§4.3 only reports qualitative results, where, according to the description, quantitative evaluation was performed
>
> Great clarification. **Appendix G.2 contains the quantitative results underlying the findings in §4.3**; we list the average number of TLCM activations across \~225 tokens. To highlight the statistical significance of this result, we’ve added the standard errors for each token (which are around \~0.2).
>
> >individual studies §4.2 and §4.3 are only done on a single model, making it difficult to understand the generalizability…
>
> Thanks for this point. To address your suggestion, **we’ve run both of these experiments on new models.**
>
> Due to limited checkpoint availability, we re-run our training experiment (4.2) on OLMo-2 7B, which features a new data mix and updated architecture. First signs of TLCM appear later, around 2T tokens, and TLCM solidifies around 3T tokens. OLMo-2 7B is trained on a total of 4T tokens. We’ve added these plots to our appendix.
>
> Re-running experiment 4.3 on Gemma-2 2B Instruct yields **highly consistent results**. Below, we list tokens with the most and least average number of TLCM activations. The upper bound of the standard error for the following TLCM activation averages is 0.38, demonstrating strong statistical significance. Like the Llama models, the beginning of sequence token has the fewest TLCM activations by far.
>
> **10 lowest activating**
>
> challenges: 9.53
> improved: 9.52
> interactive: 9.42
> enhance: 8.97
> improve: 8.91
> explore: 8.74
> feedback: 8.70
> mitigate: 7.88
> Write: 7.00
> \<bos\>: 1.00
>
> **13 most activating**
>
> \<space\>: 17.89
> 4: 17.31
> 6: 17.09
> \\t: 16.56
> 2: 16.56
> ': 16.48
> \]: 16.39
> \<space\>\<space\>: 16.36
> 7: 16.17
> \<tab\>: 16.00
> %: 15.63
> 3: 15.61
> \\n: 15.58
>
> >Figure 3 only shows layer 8 for an unknown model, and 5 “randomly sampled tokens”
>
> Although we state in L. 127–130 that this is Llama-3 8B, we agree this should be clearer. We’ve added model identifiers throughout the relevant sections.
>
> Additionally, we include full results for this experiment (total of 135 randomly sampled tokens across 18 layers) in Appendix E, Figures A7 and A8. The results in Figure 3 are largely representative of the appendix figures.
>
> >§4.2 raises three interesting potential scenarios about when TLCM may arise, but they only evaluate a single one*
>
> The experiment in Sec 4.2 tests all three scenarios. We show:
>
> 1) *Learning-induced*: TLCM does not appear in untrained models.
> 2) *Persistent*: TLCM strengthens over training, and is not—for example—some training pathology caused by instability.
> 3) *Pretraining-derived*: TLCM emerges during pretraining, not SFT or RL.
>
> >many of the findings are purely empirical, however, there lacks… theoretical discussion about TLCM
>
> Thank you for raising this point about theoretical grounding. We prioritized empirical characterization for several reasons:
>
> 1. **Phenomenon complexity requires strong empirical foundation** Our results reveal TLCM's multifaceted nature—spanning training dynamics, information flow patterns (context experiments), sublayer interactions (MLP/attention), and potentially architecture-specific effects (Appendix A). Given this complexity, we believe empirical documentation is essential before proposing theoretical explanations that might be premature or overly narrow.
>
> 2. **Building foundations for mechanistic interpretability** We view our contribution as part of a broader scientific process: establishing a corpus of well-characterized, reproducible phenomena provides the empirical constraints necessary for developing robust theories. Just as early observations of superconductivity (1911) provided crucial data for the eventual BCS theory (1957), we hope TLCM's documentation enables both practical applications and theoretical understanding.
>
> **We aim to contribute:**
>
> 1. Discovery of TLCM—a novel, broadly-observed phenomenon that reveals unexpected layer dynamics
> 2. A set of characterization experiments (e.g., 12+ models, MLP/attention differentiation, token-level statistics, training dynamics, causal interventions, context experiments).
> 3. A Jacobian eigenbasis-based interpretability technique, which we develop and apply to TLCM to show selective subspace rejection and reinforcement.
> 4. A conceptual scaffold—the propose-and-reject hypothesis that organizes observations and suggests research directions
> 5. Actionable insights for recent results in mech interp: CLTs outperforming SAEs, misfires, and model steering difficulties.
>
> We position this work as empirical science: **establishing a robust phenomenon that challenges current understanding**.
>
> >The derivation of the “propose-and-reject” hypothesis seems rather speculative
>
> The propose-and-reject hypothesis states that a layer proposes a broad swath of possible “features,” while the following layer—equipped with an attention head to gather context—removes the features that are not relevant.
>
> We intend this to be a conceptual synthesis that is consistent with all our empirical results thus far. Specifically, the hypothesis explains:
>
> * Quantitative contextual dependency experiments.
> * TLCM corrects *only* the most recent layer, rather than correcting *all prior layers*
> * TLCM’s prevalence in early layers, which are linked to “enrichment”
> * Very low TLCM activations on beginning of sequence tokens
> * TLCM emerges when considering the ***combination*** of Attention/MLP sublayers (MLPs cannot access context without attention)
> * The causal relationship we show—where the following layer dynamically attenuates the prior layer—in Sec 5.2.
> * Sec 5.3’s Jacobian experiment demonstrating **unit** attenuation ($\\lambda \= 1$) of features
>
> With this framing, we intend to **generate future research directions**, not to close the discussion.
>
> >what does it mean to “partially oppose” in a general setting, or “reverse” other layers’ contributions? Similarly, L.52…
> >Having negative cosine similarity does not mean “erasing information”
>
> Good clarification\! Our precise claim is that adjacent layers **attenuate specific components** of the prior layer’s contribution.
>
> This is supported by **stronger evidence beyond cosine similarity**:
>
> * In Sec. 5.1, causal interventions show that scaling $\\mathbf{c}\_i$​ leads to proportional counter-scaling in $\\mathbf{c}\_{i+1}$​,
>
> * In Sec. 5.2, our Jacobian eigen-analysis reveals subspaces where **unit increases receive unit correction**.
>
> To reflect your suggestion, we’ve updated our wording to emphasize “correction” and “attenuation” rather than “erasure” or “reversal.”
>
> >L.99-101: how do we know that contribution vector similarity tells us something about “novel information” and “reinforcing existing representations”?
>
> Great question\! Our interpretation draws from the common assumption in mechanistic interpretability that contributions approximate a linear combination of interpretable features (L. 89–92). Under this view:
>
> * An orthogonal contribution roughly corresponds to introducing new features,
>
> * A positively aligned contribution roughly corresponds to reinforcing existing features.
>
> To ground this in examples: Recent work (On Layer-wise Representation Similarity by Jiang et al) has found positive cosine similarity between layers in ViTs. Other work (Residual Connections Encourage Iterative Inference by Jastrzebski et al) views each residual block as performing a gradient step in activation space, so TLCM would correspond to a too high “learning rate,” which is unexpected because pretraining should find the optimal learning rate.
>
> >L.40 How do we quantitatively measure “high context dependency” in tokens?
>
> Thanks for raising this.
>
> Quantitatively, we use **number of preceding tokens** as a proxy for contextual dependency (L. 206–211). In **Figure  2b**, TLCM activation shows a linear relationship with this proxy: every additional 1k context tokens corresponds to about one additional TLCM activation, on average. While computed across 2,000 wikitext documents, we agree that this is a limited proxy.
>
> Qualitatively, we find high TLCM activation in tokens (newlines, numbers) known from prior work to rely on context. For example, newline tokens, as discussed in Lindsey et al. \[4\], often trigger planning behavior that integrates surrounding information.
>
> >lack of more rigorous derivation/description why choosing exactly for the threshold for TLCM…Looking at Figure 2b) this is not even at the low point of the distribution
>
> Good question\! Figure 2b is computed across 3.5 million adjacent layer contributions across \~100K tokens, making the **bimodality** of the cosine similarities significant. This is strong evidence that there are (at least) two distinct interaction regimes: TLCM and non-TLCM.  We agree these details are important to highlight in the paper, so we’ve added them around L. 130\.
>
> Because the $\\operatorname{cossim} \< \-0.1$ threshold is further left from the low point of the distribution, we believe it captures greater specificity than using the low point. In practice, more conservative thresholds (e.g., $\< \-0.18$) yield consistent results, as most plots already involve stronger TLCM activations due to the distribution shape.
>
> >why \[are we\] surprised by the empirical findings (elaborations of L. 99-105)
>
> Great question\! See L. 131–141 and Appendix B.1–B.2, where we explore both the theoretical and empirical significance of TLCM. Using two different sets of assumptions, we show that TLCM is highly statistically significant.
>
> >L.115-117: the authors compute the cosine similarity...but report on “correlation” in L. 116 and throughout
>
> We intended these terms to refer to the same quantity. However, we recognize your point and have updated the paper to use “cosine similarity.”
>
> >Typos, etc
>
> Thanks for pointing these out\! We have fixed these and will improve clarity for the camera-ready version.

---

> > ### Comment · Reviewer_ucnA · 2025-08-05
> >
> > I appreciate the thorough rebuttal and additional experiments.
> >
> > >Appendix G.2 contains the quantitative results underlying the findings in §4.3
> >
> > I think this is by no means a satisfying quantitative result. This is simply an even longer listing of tokens (alongside a number chosen by a rather arbitrary threshold), where a reader is expected to make qualitative judgments about commonalities between tokens with a similar number of corrections. Similarly, with the experiments shown for Gemma, the lookup table you provide is not a quantitative result that this behavior is consistent across such tokens or models – how can you claim that this is a highly consistent result from 10 sampled tokens and no formal definition of the difference between the two classes? Given that the number of corrections would be a rigorously defined measure, I would, at least expect a plot with the “context dependency” vs number of corrections, as well as a plot where this is validated on overlapping tokens across models. How would that look like?
> >
> > > Quantitatively, we use number of preceding tokens as a proxy for contextual dependency (L. 206–211)
> >
> > From the outside, the token position within a *corpus* is a weak proxy for context dependency. A token may appear at the start of a new, detached sentence, and be non-context dependent with a large token position, right? The fact that it correlates with TLCM is not a justification for that it is a good proxy of context dependency. If you want to maintain the argument of high contextual dependency, more justification is needed for me personally, even beyond the reference around newline tokens.
> >
> > > The experiment in Sec 4.2 tests all three scenarios. [...] TLCM emerges during pretraining, not SFT or RL.
> >
> > I assume §4.2 refers to figure A9 (where you, in my opinion exactly evaluate SFT), where it is visible that this effect appears for *one model* during pretraining. Did you evaluate other untrained models? (“TLCM does not appear in untrained models.” seems like too strong of a statement given the experiment. Further, did you evaluate that the TLCM effect is not reinforced during RLHF? In my opinion, you can’t make general statements like this without more evidence, and this whole section would require its own, individual study.
> >
> > >We prioritized empirical characterization… establishing a robust phenomenon
> >
> > I am not opposed to empirical characterization, however, in my opinion, the paper aims to do too much and overstates the uncovered phenomena without being able to prove many of its statements through rigorous ablations or agreed-upon metrics, as described in my original review. My suggestion is that a theoretic backing would help to solidify some of the claims made.
> >
> > >Our interpretation draws from the common assumption in mechanistic interpretability that contributions approximate a linear combination of interpretable features (L. 89–92).
> >
> > Won’t the assumptions that you mention here only hold for actual sparse autoencoders, but none of the evaluated models are SAEs, they are dense transformers..

---

> > > ### Author Response · Authors · 2025-08-08
> > >
> > > Thank you for taking the time to write a thoughtful response.
> > >
> > > >“TLCM does not appear in untrained models” seems like too strong of a statement given the experiment
> > >
> > > In Figure A9, we show training checkpoints at 5%, 20%, 66%, and 100% of pretraining. TLCM’s traces are not present at 5%, barely present at 20%, and emerge at 66%.
> > >
> > > In our rebuttal above, we perform the same experiment on a second model, finding similar results. (We cannot paste links to results).
> > >
> > > This provides evidence that TLCM does not appear in an untrained transformer. **We’re curious to understand why you believe this is too strong of a statement.**
> > >
> > > >it is visible that this effect appears for one model during pretraining
> > >
> > > To clarify, this is run on two different models (see rebuttal). This is limited by the lack of available LLM training checkpoints.
> > >
> > > With this experiment, we aimed to understand when TLCM emerges and confirm that TLCM is not an artifact of training instability. This evidence—consistency across models and the fact that TLCM is found across fully-trained models—supports our conclusions.
> > >
> > > >none of the evaluated models are SAEs, they are dense transformers
> > >
> > > In mechanistic interpretability, the superposition hypothesis refers to the idea that neural network activations are assumed to be linear combinations of “features,” which often represent semantically-meaningful concepts. Assuming the superposition hypothesis, a technique to decompose such features is an SAE. (See “Towards Monosemanticity” by Anthropic)
> > >
> > > In other words, SAEs are ***applied to*** dense transformers' activations because (1) we assume the superposition hypothesis and (2) believe that layers contribute new features to the residual stream.
> > >
> > > Our work provides **direct evidence against (2)** and implies we should use cross-layer transcoders instead (see L. 348-355). **Recent successful work employing cross-layer transcoders instead of SAEs validates our findings.**
> > >
> > > >the token position within a corpus is a weak proxy for context dependency
> > >
> > > We agree and have included this caveat in our paper. However, among proxies available in current research, we believe that token position is best for what we aim to capture. Any particular edge case is overshadowed by our averaging over thousands of documents.
> > >
> > > Ideally, we should use a standard metric to capture “contextual dependency,” the amount of information flow to a particular token. Unfortunately, no standard metric exists:
> > >
> > > 1. Recent work (Sarti et al, Cohen-Wang et al) quantified contextual dependency as the change in the token’s output distribution given some context. This does not capture our above metric. For tokens that are easy to predict (like punctuation), LLMs have been shown to use these FLOPs to aggregate information and plan (L. 195). The output distribution does not change, but information was moved into this token, increasing “contextual dependency.”
> > > 2. Other work has tried to quantify information transfer using attention scores. This has drawbacks. For example, attention scores are extremely high for BOS tokens, but research has found they are actually no-ops (see attention sinks).
> > >
> > > >how can you claim that this is a highly consistent result
> > >
> > > In Sec. 4.3, we claim that higher TLCM correction tokens include numerical tokens, date-related tokens, and punctuation tokens, while low TLCM correction tokens include the beginning of sequence token and standard English tokens. In our rebuttal, we find the same results on Gemma-2 2B Instruct. Due to space limitations, we listed only 10 tokens.
> > >
> > > In this sense, the results from the two models are consistent: **two separate models independently learn and activate the same mechanism in similar circumstances**.
> > >
> > > We discuss our intuitions around contextual dependency to foreshadow and motivate our more striking results in Section 5\. This is why we felt it useful to include.
> > >
> > > ***You raise a fair point about clarity***. We should distinguish between clear claims substantiated by strong evidence and intuitions that motivate our experiments. We’ve done this by moving our concrete claims into the “Results” section and out of the “Discussion.”
> > >
> > > >overstates the uncovered phenomena without being able to prove
> > >
> > > In our prior rebuttal, we address your concerns around vector cosine similarity, definitions of TLCM, why TLCM is surprising, low sample sizes, and more—often by pointing out detailed supporting work in the appendix.
> > >
> > > While we understand there are a couple remaining concerns, we think our most striking results (e.g. Sec 5\) are thorough—supported by appendices that validate assumptions (e.g. Appendix C). We also include initial theoretical discussions of TLCM in Appendix A. **We’re curious for your thoughts on these results as well.**
> > >
> > > We appreciate your thoughtfulness with the process. Your feedback has improved our work. However, we encourage you to consider our work in its entirety for your final review\!

---

### Official Review · Reviewer_afWX · 2025-07-03

**Clarity:** 3
**Significance:** 3
**Originality:** 3
**Rating:** 5
**Confidence:** 3

**Summary:**

The main idea of the paper is to discover a phenomenon in transformer networks, namely Transformer Layer Corrective Mechanism (TLCM) which basically means that consecutive layers in the transformer networks (e.g, in LLMs) basically correct or improve upon the representations from the previous layer. The mechanism to identify the correction is via cosine similarity in the representations obtained after each layer. Negative cosine similarity indicate that features from the previous layers are corrected by the next layer, which is evident in many of the LLMs explored in the paper.

This phenomenon is found in the early to middle layers of the LLM and is exacerbated for long context. Also seen more for numbers, dates, etc. which are more contextual in nature. Section 5 of the paper, computes TLCM when the input are perturbed to see the correction from the LLMs. Lastly, analysis from the eignevalues of the Jacobian of the transformer layers, show that the updates from the next layer are focused.

**Questions:**

1. How does this paper relate to `Layer by Layer: Uncovering Hidden Representations in Language Models` (https://arxiv.org/pdf/2502.02013)

2. Can different domains have different TLCM characteristics? (e.g, Code)

3. Could not find diagram FigureA4 in appendix corresponding to Section 4.2

**Ethical Concerns:**

["NO or VERY MINOR ethics concerns only"]

**Final Justification:**

I would like to keep my scores as the questions I raised were adequately answered. But from the discussion it seems that there are some additional details that can make this paper better.

**Limitations:**

The limitations are adequately discussed.

**Paper Formatting Concerns:**

Line 76: onn -> on.

Otherwise the paper reads well.

**Quality:**

3

**Strengths And Weaknesses:**

Strengths:

1. Overall the idea and the exploration in the paper are interesting.

2. There are empirical evidence that propose and reject hypothesis could be an interesting tool for understanding the mechanism of transformers.

Weakness:

1. While empirical evidence show that there are early signs, it would be interesting to see the results for different languages? Can this framework provide understanding on how multilingual mechanism works.

2. It could also be interesting to see the results on multimodal LLMs (e.g vision LLMs), where perturbations in images could be used to understand TLCM better.

3. While the paper explains existing interpretability challenges, future work could explore how the understanding of TLCM can be directly leveraged to develop more effective interpretability tools or even guide architectural improvements in LLMs to optimize information flow.

---

> ### Author Rebuttal · Authors · 2025-07-31
>
> Thanks for your review. We’re glad you found the “idea and exploration” interesting, the paper to “read well,” and the tools we introduce useful/interesting “for understanding the mechanisms of transformers\!”
>
> We address your questions and some of your concerns below.
>
> >It could also be interesting to see the results on multimodal LLMs (e.g vision LLMs), where perturbations in images could be used to understand TLCM better.
>
> Thanks for raising this point. Recent work (On Layer-wise Representation Similarity by Jiang et. al.) has found positive cosine similarity between layers in ViTs. However, it remains an open question whether this phenomenon is limited to just language models.
>
> >How does this paper relate to Layer by Layer: Uncovering Hidden Representations in Language Models ([https://arxiv.org/pdf/2502.02013](https://arxiv.org/pdf/2502.02013))
>
> Great question. One intuition from our work and other recent research is that:
>
> 1. Early layers are responsible for building rich representations (note: transformers have an incentive to produce representations that will be useful for future tokens, even though they cannot “see” them).
> 2. Middle-to-later layers are where final processing occurs.
> 3. Late layers focus on ensuring that the output distribution is correct, often by removing detail from the residual stream.
>
> We think this flow makes sense, and it’s also well supported by the paper you’ve linked. TLCM's high prevalence in early layers and contextual dependency gut-checks with the early layers' responsibility to “build rich representations.” Additionally, TLCM is notably less prevalent in later layers on most models, which correlates well with the less context-dependent tasks of (2) and (3).
>
> We also cite papers which build the above intuitions, see \[20, 21, 32\].
>
> >Can different domains have different TLCM characteristics? (e.g, Code) and it would be interesting to see the results for different languages
>
> Indeed, we find TLCM activations occur on code; we use code for some of our experiments (see Appendix F).
>
> Additionally, our initial experiments in Sec. 4.1 are computed across 100,000 tokens of Wikitext, which—while mostly English—includes sections in other languages. In short, we see TLCM activate across varied domains.
>
> >Could not find diagram FigureA4 in appendix corresponding to Section 4.2
>
> Good catch. We mean to refer to Figure A9. Because the paper and appendix were submitted separately at different times, it seems like we unintentionally compromised the correspondence of this figure in the paper and in the appendix. This is fixed for the camera-ready version.
>
> >TLCM can be directly leveraged to develop more effective interpretability tools or even guide architectural improvements in LLMs to optimize information flow.
>
> Great point. In Appendix A, we begin to explore the architectural features that could induce TLCM. Essentially, we introduce the concept of LayerNorm blindness, which is the phenomenon that LayerNorm causes attention and MLP sublayers to be “blind” to the norm of the residual stream. Thus, they do not know how to size their contributions and may be biased to over-contribute, so their contribution is not overshadowed. If true, this presents a real inefficiency and an opportunity to “optimize information flow.”
>
> However, we find TLCM in models that do not have LayerNorm blindness. Thus, this is not the full explanation. We agree that there is more work to do and are excited about directions like this for future work.

---

> > ### Comment · Reviewer_afWX · 2025-08-06
> >
> > The rebuttal was adequate in addressing my points. I have no further remarks and will continue monitoring the reviewer conversation.

---

### Note · Authors · 2025-08-14

We thank the reviewers and ACs for their thoughtful engagement, which has strengthened the work. Multiple reviewers described the paper as *“novel and interesting”* (dNpi, afWX, ucnA), *“striking”* with a *“good job of conducting systematic studies”* (dNpi), and *“one of the most exciting recent works in mechanistic interpretability”* (VpPo). We particularly appreciate ucnA's thorough critique, which motivated new experiments on OLMo-2 7B and Gemma-2 2B that confirmed TLCM's generalization, and improved statistical reporting throughout.

The core phenomenon remains empirically robust: across 3.5M layer pairs from 12+ models, we observe a strongly bimodal distribution establishing TLCM as a distinct operational mode. Our Section 5 results provide particularly compelling evidence—the causal intervention experiments demonstrate precise linear correction dynamics, while our Jacobian analysis reveals selective subspace targeting with eigenvalues of exactly \-1, evidence that layers perform unit correction on specific features. As VpPo noted, the intervention experiments are "impressive," showing near-perfect linear response.

We acknowledge ucnA's preference for theoretical grounding over empirical characterization. However, we maintain that discovering unexpected, robust phenomena provides the necessary empirical constraints upon which future theories can be built—e.g. superconductivity's 1911 discovery preceded BCS theory by 46 years. Our framework helps explain recent findings (CLTs\>SAEs, steering difficulties, feature non-specificity). For camera-ready, we commit to: (1) *clearly* separating empirical claims from interpretive discussion, (2) improving clarity in the Jacobian section (more detailed proof in appendix) and correction terminology, (3) adding a layer-split version of Figure 2b, and (4) other fixes as mentioned below. We believe TLCM opens important new directions for both understanding and controlling transformer-based large language models.

---

### Decision · Program_Chairs · 2025-09-17

**Decision:**

Accept (poster)

**Comment:**

This work conducts a systematic study of consecutive transformer layers, uncovering the so-called Transformer Layer Corrective Mechanism (TLCM), which boils down to adjacent layers in transformers exhibiting 'anti-correlated' activations. They observe that this phenomenon occurs across number of open-source LLMs, though is absent in models that use dropout.

Three reviewers voted for acceptance post rebuttal as they all agreed the basic thesis of the paper (TLCM) to be very interesting and worth sharing with the community:
- One reviewer found the analysis insightful and that the empirical evidence supported the claims.
- Another indicated that documented this phenomenon was novel and interesting, that the paper was easy to follow, and that the systematic study was well executed.
- The third reviewer said that this was one of the most interesting recent works in mechanistic interpretability they had read.

One reviewer remained critical because of the lack of theoretical foundation and they would have liked to see more empirical rigor, something the other reviewers did not flag. The TLCM mechanism observed and studied in this paper is different to previous studies focused on attention heads, and thus provides new insights into the internal mechanisms of transformers. I am also of the opinion that these new insights are worthwhile sharing with the community.